# FAIRGBM:
# GRADIENT BOOSTING WITH FAIRNESS CONSTRAINTS

**André F. Cruz**[1,2]    **Catarina Belém**[1,3]    **João Bravo**[1]    **Pedro Saleiro**[1]    **Pedro Bizarro**[1]

[1]Feedzai        [2]MPI for Intelligent Systems, Tübingen        [3]UC Irvine

andre.cruz@tuebingen.mpg.de    pedro.saleiro@feedzai.com

## ABSTRACT

Tabular data is prevalent in many high-stakes domains, such as financial services or public policy. Gradient Boosted Decision Trees (GBDT) are popular in these settings due to their scalability, performance, and low training cost. While fairness in these domains is a foremost concern, existing in-processing Fair ML methods are either incompatible with GBDT, or incur in significant performance losses while taking considerably longer to train. We present FairGBM, a dual ascent learning framework for training GBDT under fairness constraints, with little to no impact on predictive performance when compared to unconstrained GBDT. Since observational fairness metrics are non-differentiable, we propose smooth convex error rate proxies for common fairness criteria, enabling gradient-based optimization using a "proxy-Lagrangian" formulation. Our implementation[1] shows an order of magnitude speedup in training time relative to related work, a pivotal aspect to foster the widespread adoption of FairGBM by real-world practitioners.

## 1    INTRODUCTION

The use of Machine Learning (ML) algorithms to inform consequential decision-making has become ubiquitous in a multitude of high-stakes mission critical applications, from financial services to criminal justice or healthcare (Bartlett et al., 2019; Brennan et al., 2009; Tomar & Agarwal, 2013). At the same time, this widespread adoption of ML was followed by reports surfacing the risk of bias and discriminatory decision-making affecting people based on ethnicity, gender, age, and other sensitive attributes  (Angwin et al., 2016; Bolukbasi et al., 2016; Buolamwini & Gebru, 2018). This awareness led to the rise of Fair ML, a research area focused on discussing, measuring and mitigating the risk of bias and unfairness in ML systems. Despite the rapid pace of research in Fair ML (Hardt et al., 2016; Zafar et al., 2017; Agarwal et al., 2018; Narasimhan et al., 2019; Celis et al., 2021) and the release of several open-source software packages (Saleiro et al., 2018; Bellamy et al., 2018; Agarwal et al., 2018; Cotter et al., 2019b), there is still no clear winning method that "just works" regardless of data format and bias conditions.

Fair ML methods are usually divided into three families: pre-processing, in-processing and post-processing. Pre-processing methods aim to learn an *unbiased* representation of the training data but may not guarantee fairness in the end classifier (Zemel et al., 2013; Edwards & Storkey, 2016); while post-processing methods inevitably require test-time access to sensitive attributes and can be sub-optimal depending on the structure of the data (Hardt et al., 2016; Woodworth et al., 2017). Most in-processing Fair ML methods rely on fairness constraints to prevent the model from disproportionately hurting protected groups (Zafar et al., 2017; Agarwal et al., 2018; Cotter et al., 2019b).  Using constrained optimization, we can optimize for the predictive performance of *fair* models.

In principle, in-processing methods have the potential to introduce fairness with no training-time overhead and minimal predictive performance cost – an ideal outcome for most mission critical applications, such as financial fraud detection or medical diagnosis. Sacrificing a few percentage points of predictive performance in such settings may result in catastrophic outcomes, from safety hazards to substantial monetary losses. Therefore, the use of Fair ML in mission critical systems is particularly challenging, as fairness must be achieved with minimal performance drops.

---

[1]https://github.com/feedzai/fairgbm

Tabular data is a common data format in a variety of mission critical ML applications (*e.g.*, financial services). While deep learning is the dominant paradigm for unstructured data, gradient boosted decision trees (GBDT) algorithms are pervasive in tabular data due their state-of-the art performance and the availability of fast, scalable, ready-to-use implementations, *e.g.*, LightGBM (Ke et al., 2017) or XGBoost (Chen & Guestrin, 2016). Unfortunately, Fair ML research still lacks suitable fairness-constrained frameworks for GBDT, making it challenging to satisfy stringent fairness requirements.

As a case in point, Google's TensorFlow Constrained Optimization (TFCO) (Cotter et al., 2019b), a well-known in-processing bias mitigation technique, is only compatible with neural network models. Conversely, Microsoft's ready-to-use fairlearn EG framework (Agarwal et al., 2018) supports GBDT models, but carries a substantial training overhead, and can only output binary scores instead of a continuous scoring function, making it inapplicable to a variety of use cases. Particularly, the production of binary scores is incompatible with deployment settings with a fixed budget for positive predictions (*e.g.*, resource constraint problems (Ackermann et al., 2018)) or settings targeting a specific point in the ROC curve (*e.g.*, fixed false positive rate), such as in fraud detection.

To address this gap in Fair ML, we present FairGBM, a framework for fairness constrained optimization tailored for GBDT. Our method incorporates the classical method of Lagrange multipliers within gradient-boosting, requiring only the gradient of the constraint w.r.t. (with relation to) the model's output $\hat{Y}$. Lagrange duality enables us to perform this optimization process efficiently as a two-player game: one player minimizes the loss w.r.t. $\hat{Y}$, while the other player maximizes the loss w.r.t. the Lagrange multipliers. As fairness metrics are non-differentiable, we employ differentiable proxy constraints. Our method is inspired by the theoretical ground-work of Cotter et al. (2019b), which introduces a new "proxy-Lagrangian" formulation and proves that a stochastic equilibrium solution does exist even when employing proxy constraints. Contrary to related work, our approach does *not* require training extra models, nor keeping the training iterates in memory.

We apply our method to a real-world account opening fraud case study, as well as to five public benchmark datasets from the fairness literature (Ding et al., 2021). Moreover, we enable fairness constraint fulfillment at a specific ROC point, finding fair models that fulfill business restrictions on the number of allowed false positives or false negatives. This feature is a must for problems with high class imbalance, as the prevailing approach of using a decision threshold of $0.5$ is only optimal when maximizing accuracy. When compared with state-of-the-art in-processing fairness interventions, our method consistently achieves improved predictive performance for the same value of fairness.

In summary, this work's main contributions are:

- A novel constrained optimization framework for gradient-boosting, dubbed FairGBM.
- Differentiable proxy functions for popular fairness metrics based on the cross-entropy loss.
- A high-performance implementation[1] of our algorithm.
- Validation on a real-world case-study and five public benchmark datasets (*folktables*).

## 2 FairGBM Framework

We propose a fairness-aware variant of the gradient-boosting training framework, dubbed FairGBM. Our method minimizes predictive loss while enforcing group-wise parity on one or more error rates. We focus on the GBDT algorithm, which uses regression trees as the base weak learners (Breiman, 1984). Moreover, the current widespread use of GBDT is arguably due to two highly scalable variants of this algorithm: XGBoost (Chen & Guestrin, 2016) and LightGBM (Ke et al., 2017). In this work we provide an open-source fairness-aware implementation of LightGBM. Our work is, however, generalizable to any gradient-boosting algorithm, and to any set of differentiable constraints (not limited to fairness constraints). We refer the reader to Appendix G for notation disambiguation.

### 2.1 Optimization under Fairness Constraints

Constrained optimization (CO) approaches aim to find the set of parameters $\theta \in \Theta$ that minimize the standard predictive loss $L$ of a model $f_\theta$ given a set of $m$ fairness constraints $c_i$, $i \in \{1, ..., m\}$:

$$\theta^* = \underset{\theta \in \Theta}{\arg\min} \, L(\theta) \underset{i \in \{1,...,m\}}{\text{s.t.}} c_i(\theta) \leq 0. \tag{1}$$

This problem is often re-formulated using the Lagrangian function,

$$\mathcal{L}(\theta, \lambda) = L(\theta) + \sum_{i=1}^{m} \lambda_i c_i(\theta), \tag{2}$$

where $\lambda \in \mathbb{R}_+^m$ is the vector of Lagrange multipliers. The problem stated in Equation 1 is then, under reasonable conditions, equivalent to:

$$\theta^* = \arg\min_{\theta \in \Theta} \max_{\lambda \in \mathbb{R}_+^m} \mathcal{L}(\theta, \lambda), \tag{3}$$

which can be viewed as a zero-sum two-player game, where one player (the model player) minimizes the Lagrangian w.r.t. the model parameters $\theta$, while the other player (the constraint player) maximizes it w.r.t. the Lagrange multipliers $\lambda$ (Neumann, 1928). A pure equilibrium to this game will not exist in general for a given CO problem. Sufficient conditions for there to be one are, for example, that the original problem is a convex optimization problem satisfying an appropriate constraint qualification condition (Boyd et al., 2004). Consequently, two main issues arise when using classic CO methods with fairness metrics: the loss functions of state-of-the-art ML algorithms are non-convex (as is the case of neural networks), and standard fairness metrics are non-convex and non-differentiable.

## 2.2 Differentiable Proxies for Fairness Metrics

As a fundamentally subjective concept, there is no "one-size-fits-all" definition of fairness. Nonetheless, popular fairness notions can be defined as equalizing rate metrics across sensitive attributes (Saleiro et al., 2018; Barocas et al., 2019). For example, *equality of opportunity* (Hardt et al., 2016) is defined as equalizing expected *recall* across members of specific protected groups (*e.g.*, different genders or ethnicities). We will focus on fairness metrics for classification tasks, as the discontinuities between class membership pose a distinct challenge (non-differentiability of the step-wise function). Extension to the regression setting is fairly straightforward, as these discontinuities are no longer present and common fairness metrics are already differentiable (Agarwal et al., 2019).

In the general case, a model $f$ is deemed fair w.r.t. the sensitive attribute $S$ and some rate metric $L$ if the expected value of $L$ is independent of the value of $s \in \mathcal{S}$:

$$\mathbb{E}\left[L(f)\right] = \mathbb{E}\left[L(f)|S = s\right], \ \forall s \in \mathcal{S}. \tag{4}$$

Equation 4 can be naturally viewed as an equality constraint in the model's training process after replacing expectations under the data distribution by their sample averages. However, as discussed in Section 2.1, common fairness notions (*i.e.*, common choices of $L$) are non-convex and non-differentiable. Therefore, in order to find a solution to this CO problem, we must use some proxy metric $\tilde{L}$ that is indeed differentiable (or at least sub-differentiable) (Cotter et al., 2019b).

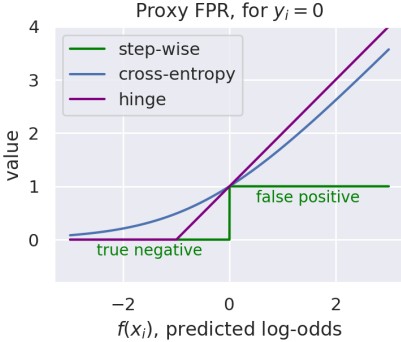

Figure 1: Convex proxies for instance-wise FPR metric, for a data sample $(x_i, y_i)$ with negative label.

Figure 1 shows examples of convex and sub-differentiable surrogates for the False Positive Rate (FPR). Equalizing FPR among sensitive attributes is also known as *predictive equality* (Corbett-Davies

| Name | Proxy metric, $\tilde{l}$ | Proxy derivative, $\frac{\partial \tilde{l}}{\partial f(x)}$ | Fairness metric |
|---|---|---|---|
| False positive | $\mathbb{I}[y = 0] \cdot \log(1 + e^{f(x)})$ | $\mathbb{I}[y = 0] \cdot \sigma(f(x))$ | predictive equality |
| False negative | $\mathbb{I}[y = 1] \cdot \log(1 + e^{-f(x)})$ | $\mathbb{I}[y = 1] \cdot [\sigma(f(x)) - 1]$ | equal opportunity |
| Predicted pos. | $\log(1 + e^{f(x)})$ | $\sigma(f(x))$ | demographic parity |
| Predicted neg. | $\log(1 + e^{-f(x)})$ | $\sigma(f(x)) - 1$ | demographic parity |

Table 1: Instance-wise metrics used to compose common error rates and corresponding cross-entropy-based proxy metrics. $\sigma$ is the sigmoid function, $f(x)$ is the predicted log-odds of instance $x$, and $y \in \{0, 1\}$ the binary label.

et al., 2017). As any function of the confusion matrix, the FPR takes in predictions binarized using a step-wise function. As no useful gradient signal can be extracted from the step-wise function, we instead use a cross-entropy-based proxy metric that upper-bounds the step-wise function. Ideally, for some fairness constraint $c$, we can guarantee its fulfillment by solving the CO problem using a proxy upper-bound $\tilde{c}$, such that $c(\theta) \leq \tilde{c}(\theta) \leq 0$. Note that, while Cotter et al. (2019b) use a hinge-based proxy, which has a discontinuous derivative, we opt for a cross-entropy-based proxy, which has a continuous derivative, leading to a smoother optimization process. Table 1 shows instance-wise rate metrics commonly used to compose fairness metrics and the proposed proxy counterparts.

In practice, the fairness constraint in Equation 4 is implemented using the set of $m = |\mathcal{S}|$ inequalities in Equation 5, *i.e.*, we have, for every $b \in \mathcal{S}$:

$$\tilde{c}_b(f) = \max_{a \in \mathcal{S}} \tilde{L}_{(S=a)}(f) - \tilde{L}_{(S=b)}(f) \leq \epsilon, \tag{5}$$

where $\epsilon \in \mathbb{R}_+ \cup \{0\}$ is the allowed constraint violation, and

$$\tilde{L}_{(S=s)}(f) = \frac{1}{\left|D_{(S=s)}\right|} \sum_{(x,y) \in D_{(S=s)}} \tilde{l}(y, f(x)), \tag{6}$$

is the proxy loss measured over the dataset $D_{(S=s)} \subseteq D$ of samples with sensitive attribute $S = s$. Original (non-proxy) counterpart functions, $c_b$ and $L_{(S=s)}$, are obtained by substituting the proxy instance-wise metric $\tilde{l}$ with its original (potentially non-differentiable) counterpart $l$.

### 2.3 FAIRNESS-AWARE GBDT

If our objective function and constraints were convex, we could find the pure Nash equilibrium of the zero sum, two-player game corresponding to the saddle point of the Lagrangian, $\mathcal{L}$. This equilibrium could be found by iterative and interleaved steps of gradient descent over our model function, $f$, and ascent over the Lagrange multipliers, $\lambda$. Importantly, this setting is relevant for GBDT models but not for Neural Networks, as the first have a convex objective and the latter do not. See Appendix C for a discussion on the limitations of our method.

However, as discussed in Section 2.2, fairness constraints are not differentiable, and we must employ differentiable proxies to use gradient-based optimization. Instead of using the Lagrangian $\mathcal{L}$, we instead use a proxy-Lagrangian $\tilde{\mathcal{L}}$,

$$\tilde{\mathcal{L}}(f, \lambda) = L(f) + \sum_{i=1}^{m} \lambda_i \tilde{c}_i(f), \tag{7}$$

where $L$ is a predictive loss function, and $\tilde{c}_i$ is a proxy inequality constraint given by Equation 5. On the other hand, simply using $\tilde{\mathcal{L}}$ for both descent and ascent optimization steps would now be enforcing our proxy-constraints and not necessarily the original ones. Thus, following Cotter et al. (2019b), we adopt a non-zero sum two-player game formulation where the descent step for the model player uses the proxy-Lagrangian $\tilde{\mathcal{L}}$ and the ascent step for the $\lambda$-player uses the Lagrangian $\mathcal{L}$ with the original constraints. The FairGBM training process (Algorithm 1) is as follows:

**Descent step.** The FairGBM descent step consists in minimizing the loss $\tilde{\mathcal{L}}$ over the function space $\mathcal{H}$ (Equation 7). That is, fitting a regression tree on the pseudo-residuals $r_{t,i} = -g_{t,i}$, where $g$ is the

---

**Algorithm 1** *FairGBM* training pseudocode

---

**Input:** $T \in \mathbb{N}$, number of boosting rounds
$\qquad \mathcal{L}, \tilde{\mathcal{L}} : \mathcal{F} \times \mathbb{R}_+^m \to \mathbb{R}$, Lagrangian and proxy-Lagrangian
$\qquad \eta_f, \eta_\lambda \in \mathbb{R}_+$, learning rates

1: Let $h_0 = \arg\min_{\gamma \in \mathbb{R}} \tilde{\mathcal{L}}(\gamma, 0)$  $\qquad\qquad\qquad\qquad\qquad$ ▷ Initial constant "guess"
2: Initialize $f \leftarrow h_0$
3: Initialize $\lambda \leftarrow 0$
4: **for** $t \in \{1, \ldots, T\}$ **do**
5: $\qquad$ Let $g_i = \frac{\partial \tilde{\mathcal{L}}(f(x_i), \lambda)}{\partial f(x_i)}$  $\qquad\qquad\qquad$ ▷ Gradient of proxy-Lagrangian w.r.t. model
6: $\qquad$ Let $\Delta = \frac{\partial \mathcal{L}(f(x_i), \lambda)}{\partial \lambda}$  $\qquad\qquad\qquad$ ▷ Gradient of Lagrangian w.r.t. multipliers
7: $\qquad$ Let $h_t = \arg\min_{h_t \in \mathcal{H}} \sum_{i=1}^{N} (-g_i - h_t(x_i))^2$  $\qquad\qquad$ ▷ Fit base learner
8: $\qquad$ Update $f \leftarrow f + \eta_f h_t$  $\qquad\qquad\qquad\qquad\qquad$ ▷ Gradient descent
9: $\qquad$ Update $\lambda \leftarrow (\lambda + \eta_\lambda \Delta)_+$  $\qquad\qquad\qquad$ ▷ Projected gradient ascent
10: **return** $h_0, \ldots, h_T$

---

gradient of the proxy-Lagrangian, $g_{t,i} = \frac{\partial \tilde{\mathcal{L}}(f, \lambda)}{\partial f(x_i)}$,

$$
g_{t,i} = \begin{cases} \frac{\partial L}{\partial f(x_i)} + (m-1) \frac{\partial \tilde{L}_{(S=j)}}{\partial f(x_i)} \sum_{k \in [m] \setminus \{j\}} \lambda_k & \text{if } s_i = j \\ \frac{\partial L}{\partial f(x_i)} - \lambda_k \frac{\partial \tilde{L}_{(S=k)}}{\partial f(x_i)} & \text{if } s_i = k \neq j \end{cases} \tag{8}
$$

where $f(x_i) = f_{t-1}(x_i)$, and $j = \arg\max_{s \in \mathcal{S}} \tilde{L}_{(S=s)}(f)$ is the group with maximal proxy loss.

**Ascent step.** The FairGBM ascent step consists in maximizing the (original) Lagrangian $\mathcal{L}$ over the multipliers $\lambda \in \Lambda$ (Equation 2). Thus, each multiplier is updated by a simple gradient ascent step:

$$
\begin{aligned}
\lambda_{t,i} &= \lambda_{t-1,i} + \eta_\lambda \frac{\partial \mathcal{L}}{\partial \lambda_i} \\
&= \lambda_{t-1,i} + \eta_\lambda c_i(f)
\end{aligned} \tag{9}
$$

where $i \in \{1, \ldots, m\}$, $m$ is the total number of inequality constraints, and $\eta_\lambda \in \mathbb{R}_+$ is the Lagrange multipliers' learning rate.

### 2.3.1 RANDOMIZED CLASSIFIER

The aforementioned FairGBM training process (Algorithm 1) converges to an approximately feasible and approximately optimal solution with known bounds to the original CO problem, dubbed a "coarse-correlated equilibrium" (Cotter et al., 2019b). This solution corresponds to a mixed strategy for the model player, defined as a distribution over all $f_t$ iterates, $t \in [1, T]$. That is, for each input $x$, we first randomly sample $t \in [1, T]$, and then use $f_t$ to make the prediction for $x$, where $f_t = \sum_{m=0}^{t} \eta_f h_m$.

In practice, using solely the last iterate $f_T$ will result in a deterministic classifier that often achieves similar metrics as the randomized classifier (Narasimhan et al., 2019), although it does not benefit from the same theoretical guarantees (Appendix E goes into further detail on this comparison). There are also several methods in the literature for reducing a randomized classifier to an approximate deterministic one (Cotter et al., 2019a).

In the general case, using this randomized classifier implies sequentially training $T$ separate models (as performed by the EG method (Agarwal et al., 2018)), severely increasing training time (by a factor of at least $T$). When using an iterative training process (such as gradient descent), it only implies training a single model, but maintaining all $T$ iterates (as performed by the TFCO method (Cotter et al., 2019b)), severely increasing memory consumption. Crucially, when using gradient boosting, each iterate contains all previous iterates. Therefore, a GBDT randomized classifier can be fully defined by maintaining solely the last iterate, carrying no extra memory consumption nor significant extra training time when compared with a vanilla GBDT classifier.

To summarize, FairGBM is the result of employing the proxy Lagrangian CO method with cross-entropy-based proxies of fairness constraints, resulting in an efficient randomized classifier with known optimality and feasibility bounds.

## 3 EXPERIMENTS

We implemented FairGBM[1] as a fork from the open-source Microsoft LightGBM implementation. The LightGBM algorithm (Ke et al., 2017) is a widely popular high performance GBDT implementation in C++, with a high-level Python interface for ease-of-use. This algorithm in particular builds on top of the standard GBDT framework by introducing *gradient-based one-side sampling* (GOSS) and *exclusive feature bundling*, both aimed at decreasing training and inference time. Although the FairGBM framework (Algorithm 1) could be applied to any gradient boosting algorithm, we choose to implement it on top of LightGBM due to its excellent scalability. Additionally, although our experiments focus on binary sensitive attributes, FairGBM can handle multiple sub-groups.

We validate our method on five large-scale public benchmark datasets, popularly known as *folktables* datasets, as well as on a real-world financial services case-study. While the *folktables* datasets provide an easily reproducible setting under common literature objectives and constraints, the real-world scenario poses a distinct set of challenges that are seldom discussed in the fairness literature, from highly imbalanced data to tight constraints on the maximum number of positive predictions.

We compare FairGBM with a set of constrained optimization baselines from the Fair ML literature. Fairlearn *EG* (Agarwal et al., 2018) is a state-of-the-art method based on the reduction of CO to a cost-sensitive learning problem. It produces a randomized binary classifier composed of several base classifiers. Fairlearn *GS* (Agarwal et al., 2018) is a similar method that instead uses a grid search over the constraint multipliers $\lambda$, and outputs a single (deterministic) classifier that achieves the best fairness-performance trade-off. *RS Reweighing* is a variation of GS that instead casts the choice of multipliers $\lambda$ as another model hyperparameter, which will be selected via Random Search (RS) — this should increase variability of trade-offs. Both EG, GS and RS are trained using LightGBM as the base algorithm, and both are implemented in the popular open-source *fairlearn* package (Bird et al., 2020). Finally, we also show results for the standard unconstrained LightGBM algorithm.

It is well-known that the choice of hyperparameters affects both performance and fairness of trained models (Cruz et al., 2021). To control for the variability of results when selecting different hyperparameters, we randomly sample 100 hyperparameter configurations of each algorithm. In the case of EG and GS, both algorithms already fit $n$ base estimators as part of a single training procedure. Hence, we run 10 trials of EG and GS, each with a budget of $n = 10$ iterations, for a total budget of 100 models trained (leading to an equal budget for all algorithms). To calculate the statistical mean and variance of each algorithm, we perform bootstrapping over the trained models (Efron & Tibshirani, 1994). Each bootstrap trial consists of $k = 20$ random draws from the pool of $n = 100$ trained models (randomly select 20%). The best performing model (the maximizer of $[\alpha \cdot performance + (1 - \alpha) \cdot fairness]$) of each trial on validation data is then separately selected. This process was repeated for 1000 trials to obtain both variance (Tables 2 and A2–A6) and confidence intervals (Figures 2b, A2b and A1). All experiments (both FairGBM and baselines) can be easily reproduced with the code provided in the supplementary materials[4].

### 3.1 DATASETS

The *folktables* datasets were put forth by Ding et al. (2021) and are derived from the American Community Survey (ACS) public use microdata sample from 2018. Each of the five datasets poses a distinct prediction task, and contains a different set of demographic features (*e.g.*, , age, marital status, education, occupation, race, gender). Notably, the ACSIncome dataset (1.6M rows) is a recreated modern-day version of the popular 1994 UCI-Adult dataset (Dua & Graff, 2017) (50K rows), which has been widely used in ML research papers over the years. On this task, the goal (label) is to predict whether US working adults' yearly income is above $50K. Due to space constraints, we focus on empirical results on the ACSIncome dataset, while the remaining four *folktables* datasets are analyzed in Appendix A. We use the same performance and fairness metrics for all *folktables* datasets: maximizing global accuracy, while equalizing group false negative rate (FNR) over different binary

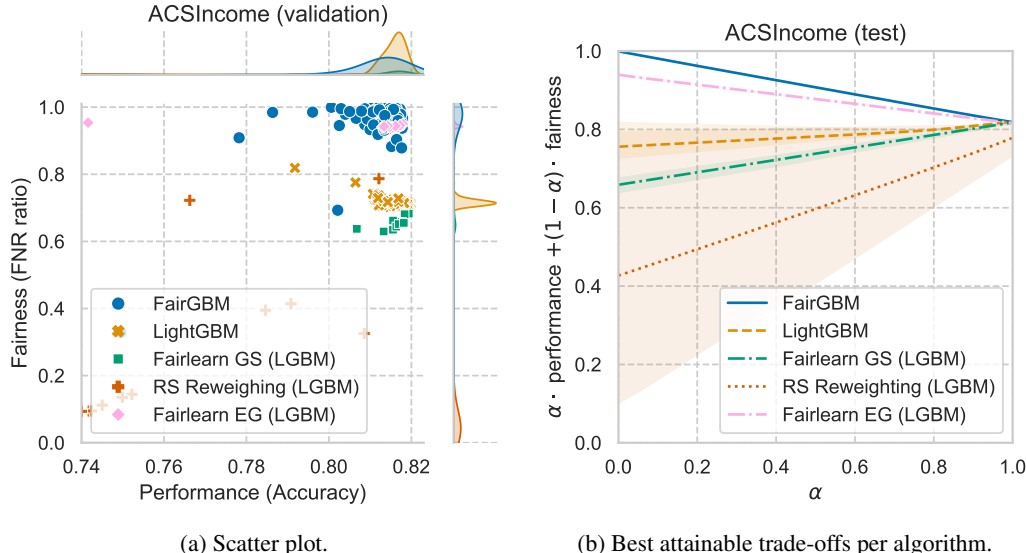

(a) Scatter plot.

(b) Best attainable trade-offs per algorithm.

Figure 2: [ACSIncome] *Left*: scatter plot showing fairness and performance of 100 trained models of each algorithm, evaluated on validation data. EG and GS show only 10 markers, as each run already trains 10 models itself. *Right*: plot of best test-set fairness-accuracy trade-offs per algorithm (models selected on validation data). Lines show the mean value, and shades show 95% confidence intervals. FairGBM (blue) achieves a statistically significant superior trade-off for all $\alpha \in [0.00, 0.99]$.

gender groups, also known as equality of opportunity (Hardt et al., 2016). Each task is randomly split in training ($60\%$), validation ($20\%$), and test ($20\%$) data.

The Account Opening Fraud (AOF) dataset, our real-world case-study, spans an 8-month period of (anonymized) data collection, containing over 500K instances. Specifically, pertaining to an online bank account application form, which also grants access to a credit line. As fraudsters are in the minority relative to legitimate applicants, our data is highly imbalanced, with only 1% fraud prevalence. This poses a distinct set of challenges and requirements for model evaluation. For example, as 99% accuracy can be trivially achieved by a constant classifier that predicts the negative class, the target performance metric is not accuracy but true positive rate (TPR) at a given false positive rate (FPR). In the AOF case, a business requirement dictates that the model must not wrongly block more than 5% of legitimate customers, *i.e.*, maximum 5% FPR. This type of requirement is arguably commonplace in production ML systems (Ackermann et al., 2018; Jesus et al., 2022). See Appendix D for details on how we operate FairGBM at arbitrary ROC points. Moreover, for the AOF case we target FPR equality among individuals of different age-groups (preventing ageism). As this task is punitive (a positive prediction leads to a negative outcome — denied account opening), a model is considered unfair if it disproportionately blocks legitimate customers of a specific protected group (given by that group's FPR). Further details on the AOF dataset are provided in Appendix B.

## 3.2 RESULTS ON THE *folktables* DATASETS

Figure 2a shows a scatter plot of the fairness-accuracy results in the validation set for models trained on the ACSIncome dataset. Note that the $x$ axis spans a small accuracy range, as all models consistently achieve high performance on this task. Figure 2b shows a plot of the best attainable trade-off for each model type (results obtained with bootstrapping as previously described). Importantly, for all trade-off choices $\alpha \in [0.00, 0.99]$, the FairGBM algorithm dominates all other methods on the scalarized metric. Only when disregarding fairness completely ($\alpha = 1.0$), do LightGBM, EG, and GS achieve similar results to FairGBM (differences within statistical insignificance). The RS algorithm suffers from severe lack of consistency, with most models being extremely unfair.

Table 2 shows another view over the same underlying results, but with a specific fairness-accuracy trade-off chosen ($\alpha = 0.75$ used for model selection), and displaying performance and fairness

| Algorithm | Trade-off $\alpha = 0.75$ | | | | Run-time | |
| | Validation | | Test | | Total (h) | Relative |
| | Fair. (%) | Perf. (%) | Fair. (%) | Perf. (%) | | |
|---|---|---|---|---|---|---|
| | ACSIncome dataset | | | | | |
| FairGBM | $99.5 \pm 0.83$ | $81.7 \pm 0.06$ | $99.3 \pm 0.89$ | $81.7 \pm 0.08$ | 9.9 | x2.1 |
| LightGBM | $75.0 \pm 3.41$ | $81.1 \pm 0.87$ | $74.6 \pm 3.57$ | $81.1 \pm 0.88$ | 4.6 | *baseline* |
| GS | $66.4 \pm 1.53$ | $81.8 \pm 0.14$ | $65.8 \pm 1.39$ | $81.8 \pm 0.14$ | 43.8 | x9.6 |
| RS | $41.4 \pm 26.6$ | $77.5 \pm 3.04$ | $41.5 \pm 26.5$ | $77.5 \pm 3.06$ | 37.1 | x8.1 |
| EG | $94.4 \pm 0.33$ | $81.6 \pm 0.15$ | $93.8 \pm 0.13$ | $81.6 \pm 0.17$ | 99.4 | x21.7 |
| | AOF dataset | | | | | |
| FairGBM | $89.3 \pm 4.62$ | $65.9 \pm 1.33$ | $87.5 \pm 3.36$ | $65.9 \pm 1.64$ | 3.5 | x2.4 |
| LightGBM | $58.0 \pm 9.39$ | $61.7 \pm 2.68$ | $66.6 \pm 14.9$ | $61.1 \pm 2.86$ | 1.4 | *baseline* |
| GS | $98.5 \pm 1.00$ | $23.6 \pm 3.45$ | $98.4 \pm 1.67$ | $23.7 \pm 3.64$ | 21.4 | x14.7 |
| RS | $84.0 \pm 19.3$ | $36.9 \pm 8.43$ | $84.6 \pm 20.9$ | $37.4 \pm 8.89$ | 10.3 | x7.1 |

Table 2: Mean and standard deviation of results on the ACSIncome (top rows) and AOF (bottom rows) datasets, with the model-selection trade-off set as $\alpha = 0.75$. For each row, we select the model that maximizes $[\alpha \cdot performance + (1 - \alpha) \cdot fairness]$ measured in validation, and report results on both validation and test data. See related Tables A2 and A1 for results with other trade-off choices.

results instead of the scalarized objective. Table A2 shows ACSIncome results for two other trade-off choices: $\alpha \in \{0.50, 0.95\}$. Among all tested algorithms, FairGBM has the lowest average constraint violation for all three studied values of $\alpha$ on the ACSIncome dataset ($p < 0.01$ for all pair-wise comparisons), while achieving better performance than RS, and similar performance (differences are not statistically significant) to LightGBM, GS, and EG — *i.e.*, FairGBM models are Pareto dominant over the baselines. The EG algorithm follows, also achieving high performance and high fairness, although significantly behind FairGBM on the latter ($p < 0.01$). At the same time, the GS and RS Reweighing algorithms achieve a surprisingly low fairness on this dataset, signalling that their ensembled counterpart (the EG algorithm) seems better fitted for this setting. As expected, fairness for the unconstrained LightGBM algorithm is considerably lower than that of FairGBM or EG.

A similar trend is visible on the other four *folktables* datasets (see Figure A1 and Tables A3–A6). FairGBM consistently achieves the best fairness-accuracy trade-offs among all models, either isolated (ACSIncome and ACSEmployment), tied with EG (ACSTravelTime), or tied with both EG and GS (ACSMobility and ACSPublicCoverage). Collectively, EG is arguably the strongest CO baseline, followed by GS, and then RS. However, the total time taken to train all FairGBM models is under a tenth of the time taken to train all EG models (see rightmost columns of Table 2); and EG also requires keeping tens of models in memory ($n = 10$ in our experiments), straining possible scalability.

### 3.3 RESULTS ON THE ACCOUNT OPENING FRAUD DATASET

While most models achieve high performance on the ACSIncome dataset, on AOF we see a significantly wider range of performance values (compare AOF plot in Figure A2a with ACSIncome plot in Figure 2a). Moreover, the unconstrained LightGBM algorithm in this setting shows significant average *unfairness*, achieving its peak performance at approximately 33% fairness.

On the AOF test set, FairGBM dominates LightGBM on both fairness *and* performance for the $\alpha = 0.5$ and $\alpha = 0.75$ trade-offs, while achieving superior fairness with statistically insignificant performance differences on the $\alpha = 0.95$ trade-off (results for all three trade-offs in Table A1). Remaining baselines achieve high fairness at an extreme performance cost when compared to FairGBM. For example, on $\alpha = 0.75$ (Table 2), GS achieves near perfect fairness ($98.4 \pm 1.67$) but catches only 36% of the fraud instances that FairGBM catches ($23.7/65.9 = 0.36$), while taking 6 times longer to train ($14.7/2.4 = 6.1$). In fact, FairGBM significantly extends the Pareto frontier of attainable trade-offs when compared to any other model in the comparison (see Figure A2b).

Note that the EG method was excluded from the comparison on the AOF dataset as it has critical incompatibilities with this real-world setting. Importantly, it produces a randomized binary classifier

that implicitly uses a $0.50$ decision threshold. This is optimal to maximize accuracy – which is trivial on AOF due to its extreme class imbalance – but severely sub-optimal to maximize TPR. Due to lack of real-valued score predictions, neither can we compute a score threshold after training to maximize TPR on the model's predictions, nor can we fulfill the 5% FPR business constraint. Nonetheless, EG is still part of the comparison on the five *folktables* datasets.

## 4 RELATED WORK

Prior work on algorithmic fairness can be broadly divided into three categories: pre-processing, in-processing, and post-processing; depending on whether it acts on the training data, the training process, or the model's predictions, respectively.

**Pre-processing** methods aim to modify the input data such that any model trained on it would no longer exhibit biases. This is typically achieved either by (1) creating a new representation $U$ of the features $X$ that does not exhibit correlations with the protected attribute $S$ (Zemel et al., 2013; Edwards & Storkey, 2016), or (2) by altering the label distribution $Y$ according to some heuristic (Fish et al., 2016; Kamiran & Calders, 2009) (*e.g.*, equalizing prevalence across sub-groups of the population). Although compatible with any downstream task, by acting on the beginning of the ML pipeline these methods may not be able to guarantee fairness on the end model. Moreover, recent empirical comparisons have shown that pre-processing methods often lag behind in-processing and post-processing methods (Ding et al., 2021).

**In-processing** methods alter the learning process itself in order to train models that make fairer predictions. There are a wide variety of approaches under this class of methods: training under fairness constraints (Zafar et al., 2017; Agarwal et al., 2018; Cotter et al., 2019b), using a loss function that penalizes unfairness (Fish et al., 2016; Iosifidis & Ntoutsi, 2019; Ravichandran et al., 2020), or training with an adversary that tries to predict protected-group membership (Grari et al., 2019). The main shortcoming of in-processing methods lies in their selective compatibility with particular algorithms or families of algorithms. As a case in point, there is currently no constrained optimization method tailored for the GBDT algorithm, besides the one present in this work. However, the state-of-the-art results for numerous tabular data tasks are currently held by boosting-based models (Shwartz-Ziv & Armon, 2021). AdaFair (Iosifidis & Ntoutsi, 2019) is a bias mitigation method for the AdaBoost algorithm (Freund & Schapire, 1996), a similar method to GBDT. However, we did not consider it as a direct baseline in this work as it is only compatible with the equal odds fairness metric (Hardt et al., 2016), and not with equal opportunity or predictive equality (used in our experiments). Moreover, this method lacks any theoretical guarantees, employing a series of heuristics used to change the weights of samples from underprivileged groups.

**Post-processing** methods alter the model's predictions to fulfill some statistical measure of fairness. In practice, this is done by (1) shifting the decision boundary for specific sub-groups (Hardt et al., 2016; Fish et al., 2016), or by (2) randomly classifying a portion of individuals of the underprivileged group (Kamiran et al., 2012; Pleiss et al., 2017). Methods based on shifting the decision-boundary have the clear advantage of achieving 100% fairness in the data where they are calibrated (training or validation data), while also being compatible with any score-based classifier. However, post-processing methods can be highly sub-optimal (Woodworth et al., 2017), as they act on the model *after* it was learned. Moreover, they can lead to higher performance degradation when compared to in-processing methods (Ding et al., 2021).

## 5 CONCLUSION

We presented FairGBM, a dual ascent learning framework under fairness constraints specifically tailored for gradient boosting. To enable gradient-based optimization we propose differentiable proxies for popular fairness metrics that are able to attain state-of-the-art fairness-performance trade-offs on tabular data. When compared with general-purpose constrained optimization methods, FairGBM is more consistent across datasets, and typically achieves higher fairness for the same level of performance. Crucially, FairGBM does not require significant extra computational resources, while related CO algorithms considerably increase training time and/or memory consumption. Finally, we enable fairness constraint fulfillment at a specified ROC point or with a fixed budget for positive predictions, a common requirement in real-world high-stakes settings.

ACKNOWLEDGMENTS

The authors thank Sérgio Jesus (Feedzai) for invaluable feedback and help in the paper review rebuttal process. The authors thank the International Max Planck Research School for Intelligent Systems (IMPRS-IS) for supporting André F. Cruz during part of this research.

The project CAMELOT (reference POCI-01-0247-FEDER-045915) leading to this work is co-financed by the ERDF - European Regional Development Fund through the Operational Program for Competitiveness and Internationalisation - COMPETE 2020, the North Portugal Regional Operational Program - NORTE 2020 and by the Portuguese Foundation for Science and Technology - FCT under the CMU Portugal international partnership.

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

# A  ADDITIONAL RESULTS

This section displays results on the four *folktables* datasets that were omitted from the main body: ACSEmployment, ACSMobility, ACSTravelTime, and ACSPublicCoverage (Ding et al., 2021). Each dataset poses a distinct prediction task: ACSEmployment (2.3M rows) relates to employment status prediction, ACSMobility (0.6M rows) relates to prediction of address changes, ACSTravelTime (1.4M rows) relates to prediction of the length of daily work commute, and ACSPublicCoverage (1.1M rows) relates to prediction of public health insurance coverage. Additionally, we display extra plots and results for other trade-off choices on the ACSIncome and AOF datasets.

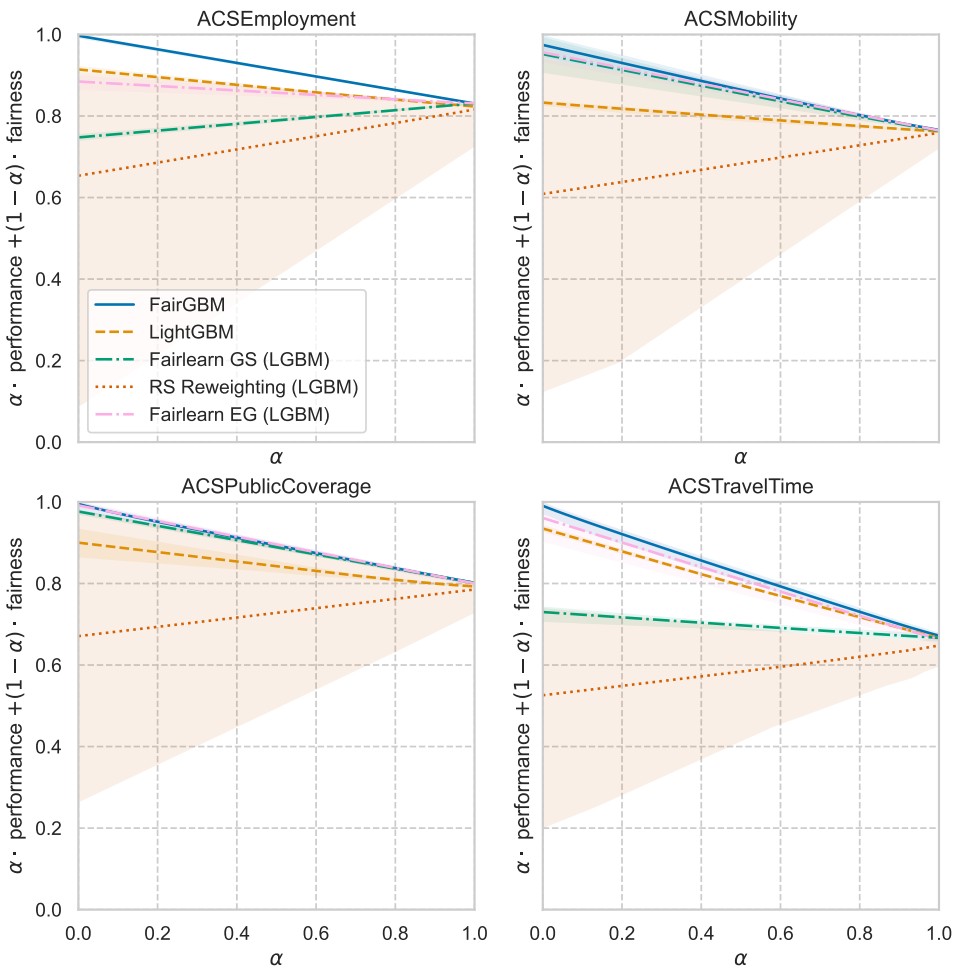

Figure A1: Plot of best test-set fairness-accuracy trade-offs per algorithm (models selected on validation data), on four *folktables* datasets. Lines show the mean value, and shades show 95% confidence intervals.

Figure A1 shows plots of the test-set $\alpha$-weighted metric attainable by each algorithm as a function of $\alpha \in [0, 1]$, on each *folktables* dataset (model-selection based on validation results). Results follow a similar trend to those seen on the ACSIncome dataset (Figure 2b): FairGBM consistently achieves the top spot, either isolated (ACSEmployment and ACSIncome) or tied with other methods on part of the trade-off spectrum (ACSMobility, ACSPublicCoverage and ACSTravelTime). EG achieves competitive trade-offs as well, although not as consistently across datasets, and at a high CPU training cost. GS ties with EG and FairGBM for the best trade-offs on the ACSMobility and ACSPublicCoverage datasets. Detailed fairness and performance results for all *folktables* datasets (for $\alpha \in \{0.50, 0.75, 0.95\}$) are shown in Tables A2–A6.

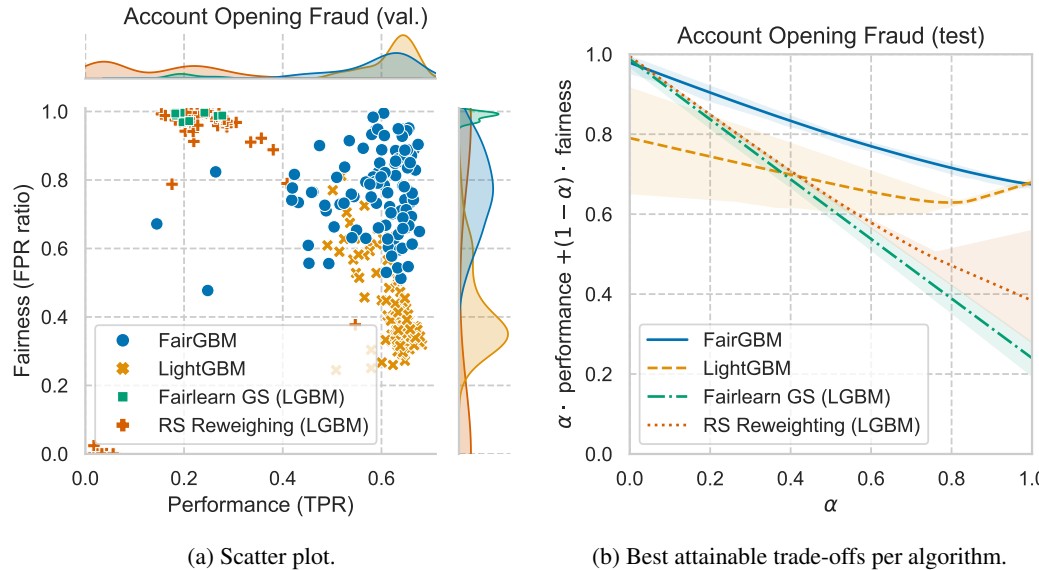

(a) Scatter plot.

(b) Best attainable trade-offs per algorithm.

Figure A2: [AOF dataset] *Left*: scatter plot showing fairness and performance of 100 trained models of each algorithm, evaluated on validation data. GS shows only 10 markers, as each run already trains 10 models itself. *Right*: plot of best test-set fairness-accuracy trade-offs per algorithm (models selected on validation data). Lines show the mean value, and shades show 95% confidence intervals. FairGBM (blue) achieves a statistically significant superior trade-off for all $\alpha \in [0.05, 0.98]$.

**Account Opening Fraud (AOF)**

| Algorithm | Validation | | Test | |
|---|---|---|---|---|
| | **Fair. (%)** | **Perf. (%)** | **Fair. (%)** | **Perf. (%)** |
| Trade-off $\alpha = 0.50$ | | | | |
| FairGBM | $92.2 \pm 4.19$ | $64.8 \pm 2.54$ | $90.1 \pm 4.51$ | $64.7 \pm 2.59$ |
| LightGBM | $70.2 \pm 7.92$ | $55.8 \pm 3.97$ | $78.4 \pm 9.89$ | $55.8 \pm 3.36$ |
| GS | $98.9 \pm 0.82$ | $23.4 \pm 3.71$ | $98.7 \pm 1.37$ | $23.4 \pm 3.87$ |
| RS | $95.3 \pm 3.55$ | $31.2 \pm 4.31$ | $96.1 \pm 3.47$ | $31.3 \pm 4.53$ |
| Trade-off $\alpha = 0.75$ | | | | |
| FairGBM | $89.3 \pm 4.62$ | $65.9 \pm 1.33$ | $87.5 \pm 3.36$ | $65.9 \pm 1.64$ |
| LightGBM | $58.0 \pm 9.39$ | $61.7 \pm 2.68$ | $66.6 \pm 14.9$ | $61.1 \pm 2.86$ |
| GS | $98.5 \pm 1.0$ | $23.6 \pm 3.45$ | $98.4 \pm 1.67$ | $23.7 \pm 3.64$ |
| RS | $84.0 \pm 19.3$ | $36.9 \pm 8.43$ | $84.6 \pm 20.9$ | $37.4 \pm 8.89$ |
| Trade-off $\alpha = 0.95$ | | | | |
| FairGBM | $80.0 \pm 9.79$ | $66.6 \pm 0.88$ | $80.2 \pm 9.13$ | $66.6 \pm 1.06$ |
| LightGBM | $33.7 \pm 1.70$ | $67.6 \pm 0.47$ | $36.0 \pm 1.29$ | $67.3 \pm 1.01$ |
| GS | $98.5 \pm 0.99$ | $23.6 \pm 3.44$ | $98.3 \pm 1.74$ | $23.8 \pm 3.51$ |
| RS | $81.5 \pm 21.2$ | $37.6 \pm 9.15$ | $82.2 \pm 23.2$ | $38.2 \pm 9.64$ |

Table A1: Mean and standard deviation of results on the AOF dataset, for three different choices of model-selection trade-off: $\alpha \in \{0.50, 0.75, 0.95\}$. Model selection metric was $[\alpha \cdot performance + (1 - \alpha) \cdot fairness]$. The best model is selected on validation data, and results are reported on both validation and test data.

Figure A2 shows a scatter plot and a plot of best attainable trade-offs on the AOF dataset. When compared to the ACSIncome results (Figure 2), we can see a significantly wider range of attainable performance and fairness values, arguably making it a more challenging but more interesting task. These differences further motivate our focus on a real-world setting.

**ACSIncome**

| Algorithm | Validation | | Test | |
|---|---|---|---|---|
| | Fair. (%) | Perf. (%) | Fair. (%) | Perf. (%) |
| Trade-off $\alpha = 0.50$ | | | | |
| FairGBM | $99.6 \pm 0.73$ | $81.6 \pm 0.08$ | $99.4 \pm 0.74$ | $81.7 \pm 0.09$ |
| LightGBM | $75.6 \pm 3.28$ | $80.8 \pm 0.80$ | $75.5 \pm 3.32$ | $80.8 \pm 0.80$ |
| GS | $66.4 \pm 1.49$ | $81.7 \pm 0.19$ | $65.9 \pm 1.36$ | $81.8 \pm 0.20$ |
| RS | $42.3 \pm 25.8$ | $77.0 \pm 3.70$ | $42.4 \pm 25.8$ | $77.0 \pm 3.73$ |
| EG | $94.4 \pm 0.32$ | $81.6 \pm 0.16$ | $93.8 \pm 0.11$ | $81.6 \pm 0.18$ |
| Trade-off $\alpha = 0.75$ | | | | |
| FairGBM | $99.5 \pm 0.83$ | $81.7 \pm 0.06$ | $99.3 \pm 0.89$ | $81.7 \pm 0.08$ |
| LightGBM | $75.0 \pm 3.41$ | $81.1 \pm 0.87$ | $74.6 \pm 3.57$ | $81.1 \pm 0.88$ |
| GS | $66.4 \pm 1.53$ | $81.8 \pm 0.14$ | $65.8 \pm 1.39$ | $81.8 \pm 0.14$ |
| RS | $41.4 \pm 26.6$ | $77.5 \pm 3.04$ | $41.5 \pm 26.5$ | $77.5 \pm 3.06$ |
| EG | $94.4 \pm 0.33$ | $81.6 \pm 0.15$ | $93.8 \pm 0.13$ | $81.6 \pm 0.17$ |
| Trade-off $\alpha = 0.95$ | | | | |
| FairGBM | $98.7 \pm 1.07$ | $81.8 \pm 0.06$ | $98.5 \pm 1.22$ | $81.8 \pm 0.06$ |
| LightGBM | $71.2 \pm 0.30$ | $81.9 \pm 0.04$ | $70.8 \pm 0.34$ | $82.0 \pm 0.05$ |
| GS | $66.3 \pm 1.54$ | $81.8 \pm 0.14$ | $65.8 \pm 1.40$ | $81.8 \pm 0.14$ |
| RS | $38.3 \pm 25.6$ | $77.8 \pm 2.96$ | $38.4 \pm 25.6$ | $77.8 \pm 2.97$ |
| EG | $94.4 \pm 0.36$ | $81.6 \pm 0.15$ | $93.9 \pm 0.13$ | $81.6 \pm 0.16$ |

Table A2: Mean and standard deviation of results on the ACSIncome dataset.

**ACSEmployment**

| Algorithm | Validation | | Test | |
|---|---|---|---|---|
| | Fair. (%) | Perf. (%) | Fair. (%) | Perf. (%) |
| Trade-off $\alpha = 0.50$ | | | | |
| FairGBM | $99.6 \pm 0.26$ | $83.1 \pm 0.07$ | $99.4 \pm 0.30$ | $83.0 \pm 0.08$ |
| LightGBM | $91.8 \pm 0.58$ | $82.1 \pm 0.22$ | $91.4 \pm 0.50$ | $82.0 \pm 0.25$ |
| GS | $73.9 \pm 0.85$ | $83.1 \pm 0.13$ | $74.7 \pm 0.84$ | $83.1 \pm 0.14$ |
| RS | $65.2 \pm 25.6$ | $81.6 \pm 2.93$ | $65.3 \pm 25.5$ | $81.5 \pm 2.95$ |
| EG | $87.6 \pm 1.11$ | $83.0 \pm 0.16$ | $88.4 \pm 1.12$ | $83.0 \pm 0.17$ |
| Trade-off $\alpha = 0.75$ | | | | |
| FairGBM | $99.6 \pm 0.29$ | $83.1 \pm 0.03$ | $99.4 \pm 0.31$ | $83.1 \pm 0.04$ |
| LightGBM | $91.4 \pm 0.55$ | $82.3 \pm 0.17$ | $91.1 \pm 0.48$ | $82.2 \pm 0.19$ |
| GS | $73.8 \pm 1.01$ | $83.2 \pm 0.06$ | $74.7 \pm 0.99$ | $83.1 \pm 0.07$ |
| RS | $65.1 \pm 25.6$ | $81.6 \pm 2.92$ | $65.2 \pm 25.5$ | $81.5 \pm 2.93$ |
| EG | $87.6 \pm 1.11$ | $83.1 \pm 0.15$ | $88.4 \pm 1.12$ | $83.0 \pm 0.16$ |
| Trade-off $\alpha = 0.95$ | | | | |
| FairGBM | $99.5 \pm 0.36$ | $83.1 \pm 0.02$ | $99.4 \pm 0.40$ | $83.1 \pm 0.02$ |
| LightGBM | $91.0 \pm 0.15$ | $82.4 \pm 0.01$ | $90.8 \pm 0.11$ | $82.3 \pm 0.01$ |
| GS | $73.7 \pm 1.00$ | $83.2 \pm 0.06$ | $74.7 \pm 0.98$ | $83.1 \pm 0.07$ |
| RS | $65.0 \pm 25.5$ | $81.6 \pm 2.93$ | $65.1 \pm 25.4$ | $81.6 \pm 2.94$ |
| EG | $86.8 \pm 1.65$ | $83.2 \pm 0.06$ | $87.7 \pm 1.67$ | $83.1 \pm 0.06$ |

Table A3: Mean and standard deviation of results on the ACSEmployment dataset.

**ACSMobility**

| Algorithm | Validation | | Test | |
|---|---|---|---|---|
| | Fair. (%) | Perf. (%) | Fair. (%) | Perf. (%) |
| Trade-off $\alpha = 0.50$ | | | | |
| FairGBM | $95.0 \pm 1.36$ | $75.8 \pm 0.74$ | $96.2 \pm 2.44$ | $75.7 \pm 0.74$ |
| LightGBM | $81.1 \pm 1.44$ | $76.1 \pm 0.31$ | $82.3 \pm 1.03$ | $76.1 \pm 0.30$ |
| GS | $92.4 \pm 2.46$ | $76.0 \pm 0.89$ | $95.0 \pm 2.92$ | $75.8 \pm 0.94$ |
| RS | $60.0 \pm 25.3$ | $75.9 \pm 2.27$ | $60.8 \pm 26.5$ | $75.8 \pm 2.26$ |
| EG | $96.3 \pm 2.50$ | $76.5 \pm 0.13$ | $94.6 \pm 2.11$ | $76.4 \pm 0.12$ |
| Trade-off $\alpha = 0.75$ | | | | |
| FairGBM | $93.9 \pm 1.59$ | $76.4 \pm 0.28$ | $94.4 \pm 1.85$ | $76.3 \pm 0.28$ |
| LightGBM | $81.0 \pm 1.50$ | $76.1 \pm 0.09$ | $82.3 \pm 1.02$ | $76.1 \pm 0.08$ |
| GS | $92.0 \pm 2.18$ | $76.2 \pm 0.68$ | $94.6 \pm 2.67$ | $76.0 \pm 0.72$ |
| RS | $60.0 \pm 25.3$ | $75.9 \pm 2.25$ | $60.8 \pm 26.5$ | $75.8 \pm 2.24$ |
| EG | $96.3 \pm 2.52$ | $76.5 \pm 0.13$ | $94.7 \pm 2.09$ | $76.4 \pm 0.11$ |
| Trade-off $\alpha = 0.95$ | | | | |
| FairGBM | $92.1 \pm 1.42$ | $76.7 \pm 0.09$ | $93.7 \pm 1.19$ | $76.5 \pm 0.08$ |
| LightGBM | $79.9 \pm 1.40$ | $76.2 \pm 0.07$ | $81.7 \pm 1.07$ | $76.2 \pm 0.06$ |
| GS | $90.8 \pm 2.32$ | $76.4 \pm 0.11$ | $93.7 \pm 2.66$ | $76.3 \pm 0.12$ |
| RS | $59.8 \pm 25.4$ | $76.0 \pm 2.22$ | $60.5 \pm 26.6$ | $75.9 \pm 2.20$ |
| EG | $96.0 \pm 2.90$ | $76.6 \pm 0.11$ | $94.6 \pm 2.02$ | $76.4 \pm 0.09$ |

Table A4: Mean and standard deviation of results on the ACSMobility dataset.

**ACSTravelTime**

| Algorithm | Validation | | Test | |
|---|---|---|---|---|
| | Fair. (%) | Perf. (%) | Fair. (%) | Perf. (%) |
| Trade-off $\alpha = 0.50$ | | | | |
| FairGBM | $98.2 \pm 1.28$ | $66.7 \pm 0.99$ | $98.3 \pm 1.39$ | $66.6 \pm 0.98$ |
| LightGBM | $93.6 \pm 0.68$ | $65.8 \pm 0.59$ | $93.1 \pm 0.59$ | $65.8 \pm 0.64$ |
| GS | $72.8 \pm 1.23$ | $66.6 \pm 0.52$ | $73.0 \pm 0.98$ | $66.5 \pm 0.52$ |
| RS | $52.5 \pm 20.6$ | $64.3 \pm 2.19$ | $52.5 \pm 20.6$ | $64.3 \pm 2.18$ |
| EG | $96.1 \pm 2.96$ | $66.0 \pm 0.81$ | $96.1 \pm 2.96$ | $66.0 \pm 0.78$ |
| Trade-off $\alpha = 0.75$ | | | | |
| FairGBM | $97.7 \pm 1.83$ | $67.0 \pm 0.71$ | $97.6 \pm 1.96$ | $66.9 \pm 0.71$ |
| LightGBM | $92.5 \pm 0.42$ | $66.5 \pm 0.24$ | $92.4 \pm 0.28$ | $66.6 \pm 0.24$ |
| GS | $72.6 \pm 1.26$ | $66.7 \pm 0.49$ | $72.8 \pm 1.01$ | $66.6 \pm 0.50$ |
| RS | $52.1 \pm 21.0$ | $64.6 \pm 1.85$ | $52.1 \pm 21.0$ | $64.5 \pm 1.84$ |
| EG | $95.7 \pm 3.06$ | $66.2 \pm 0.82$ | $95.7 \pm 3.05$ | $66.2 \pm 0.79$ |
| Trade-off $\alpha = 0.95$ | | | | |
| FairGBM | $95.6 \pm 4.72$ | $67.2 \pm 0.50$ | $95.5 \pm 4.64$ | $67.2 \pm 0.52$ |
| LightGBM | $92.3 \pm 0.33$ | $66.6 \pm 0.22$ | $92.3 \pm 0.27$ | $66.6 \pm 0.21$ |
| GS | $72.3 \pm 1.33$ | $66.7 \pm 0.45$ | $72.6 \pm 1.07$ | $66.7 \pm 0.46$ |
| RS | $49.6 \pm 20.9$ | $64.8 \pm 1.81$ | $49.6 \pm 21.0$ | $64.7 \pm 1.82$ |
| EG | $94.3 \pm 4.12$ | $66.4 \pm 0.77$ | $94.2 \pm 4.18$ | $66.4 \pm 0.74$ |

Table A5: Mean and standard deviation of results on the ACSTravelTime dataset.

**ACSPublicCoverage**

| Algorithm | Validation | | Test | |
|---|---|---|---|---|
| | Fair. (%) | Perf. (%) | Fair. (%) | Perf. (%) |
| Trade-off $\alpha = 0.50$ | | | | |
| FairGBM | $99.7 \pm 0.47$ | $79.9 \pm 0.23$ | $98.0 \pm 0.65$ | $80.0 \pm 0.26$ |
| LightGBM | $89.1 \pm 2.04$ | $78.4 \pm 0.58$ | $90.0 \pm 2.05$ | $78.5 \pm 0.63$ |
| GS | $96.2 \pm 0.35$ | $79.9 \pm 0.12$ | $97.6 \pm 0.30$ | $80.1 \pm 0.15$ |
| RS | $66.6 \pm 24.8$ | $78.4 \pm 2.21$ | $67.1 \pm 25.7$ | $78.5 \pm 2.25$ |
| EG | $98.7 \pm 1.18$ | $79.9 \pm 0.23$ | $98.9 \pm 0.88$ | $80.0 \pm 0.23$ |
| Trade-off $\alpha = 0.75$ | | | | |
| FairGBM | $99.5 \pm 0.56$ | $80.0 \pm 0.17$ | $97.9 \pm 0.74$ | $80.1 \pm 0.19$ |
| LightGBM | $88.5 \pm 1.62$ | $78.6 \pm 0.41$ | $89.4 \pm 1.63$ | $78.7 \pm 0.45$ |
| GS | $96.1 \pm 0.40$ | $80.0 \pm 0.11$ | $97.7 \pm 0.26$ | $80.1 \pm 0.15$ |
| RS | $66.5 \pm 24.9$ | $78.4 \pm 2.18$ | $67.1 \pm 25.7$ | $78.5 \pm 2.21$ |
| EG | $98.6 \pm 1.30$ | $79.9 \pm 0.19$ | $98.9 \pm 0.93$ | $80.0 \pm 0.19$ |
| Trade-off $\alpha = 0.95$ | | | | |
| FairGBM | $99.0 \pm 0.96$ | $80.1 \pm 0.10$ | $97.7 \pm 1.24$ | $80.2 \pm 0.11$ |
| LightGBM | $85.5 \pm 1.22$ | $79.1 \pm 0.08$ | $86.4 \pm 1.23$ | $79.3 \pm 0.08$ |
| GS | $96.0 \pm 0.40$ | $80.0 \pm 0.12$ | $97.6 \pm 0.26$ | $80.1 \pm 0.15$ |
| RS | $66.5 \pm 24.9$ | $78.5 \pm 2.17$ | $67.0 \pm 25.7$ | $78.5 \pm 2.20$ |
| EG | $98.4 \pm 1.42$ | $79.9 \pm 0.15$ | $98.8 \pm 1.06$ | $80.0 \pm 0.16$ |

Table A6: Mean and standard deviation of results on the ACSPublicCoverage dataset.

## B  DESCRIPTION OF ACCOUNT OPENING FRAUD DATASET

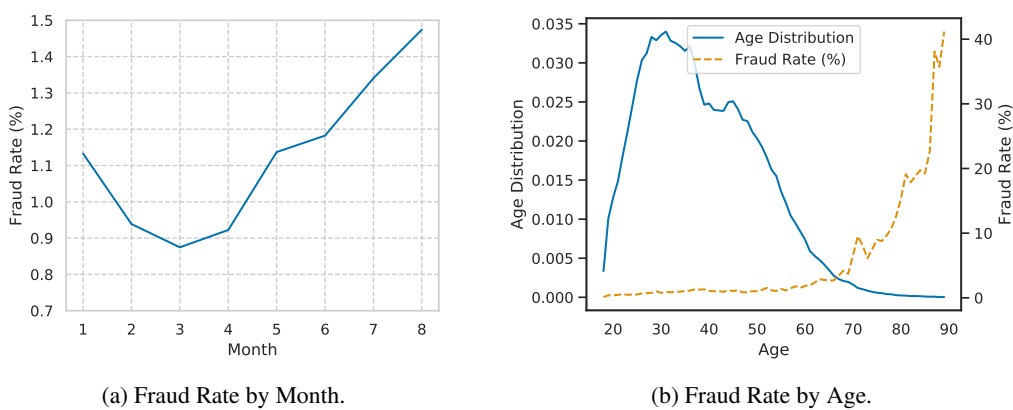

(a) Fraud Rate by Month.  (b) Fraud Rate by Age.

Figure A3: Plot of the variation of Fraud Rate depending on the month of the application and applicant's age. The plot also contains distribution of Age over all the applications in the dataset.

The Account Opening Fraud (AOF) dataset is an in-house dataset that comprises 8 months of data from a real-world fraud detection task. Specifically, on the detection of fraudulent online bank account opening applications in a large European consumer bank. In this setting, fraudsters attempt to either impersonate someone via identity theft, or create a fictional individual in order to be approved for a new bank account. After being granted access to a new bank account, the fraudster quickly maxes out the line of credit, or uses the account to receive illicit payments. All costs are sustained by the bank.

The temporal aspect of the dataset plays an important role, as oftentimes fraudsters adapt their strategies over time to try to improve their success rate. This translates into considerable concept

drift throughout the year (*e.g.*, a model trained on 1-year-old data will perform poorly on recent data). With this in mind, we split the AOF dataset temporally, using 6 months for training, 1 month for validation, and 1 month for test, such that we train on the oldest data and test on the most recent. The simpler strategy of randomly splitting in train/test/validation in this setting would not properly mimic a real-world environment, and would lead to over-optimistic results.

Each instance (row) of the dataset represents an individual application. All of the applications were made on an online platform, where explicit consent to store and process the gathered data was granted by the applicant. Each instance is labeled after a few months of activity, as by then it is apparent whether the account owner is fraudulent (positive label) or legitimate (negative label). A total of 67 features are stored for each applicant, 6 being categorical, 19 binary, and 42 numerical. Most features represent aggregations over the original information and context of the application (*e.g.*, count of the number of applications in the last hour).

The prevalence of fraud varies between $0.85\%$ and $1.5\%$ during the eight month period (see Figure A3a). We observe that these values are higher for the later months, which were used as validation (the $7^{th}$, or second-to-last month) and test (the $8^{th}$, or last month) data. Additionally, the distribution of applications also changes from month to month, ranging from $9.5\%$ on the lower end to $15\%$ on the higher end. The extremely high class imbalance (approximately 1 positive label for each 99 negative labels), the gradual change in fraud prevalence, together with naturally shifting consumer patterns along 8 months of real-world data are examples of common real-world challenges that are not often present in datasets from the fairness literature.

Due to the low fraud rate in the data, measuring accuracy of the models would lead to misleading results. A trivial classifier that constantly outputs legitimate predictions would achieve an accuracy close to $99\%$. To address this, a commonly used performance metric in the fraud detection industry is the true positive rate (TPR, or Recall), as it reflects the percentage of fraud that was caught by the ML model. Moreover, in order to keep attrition on legitimate customers low, a requirement of at most 5% false positive rate (FPR) is used, *i.e.*, at most 5% of legitimate (label negative) customers are wrongly blocked from opening a bank account. Additionally, due to the punitive nature of the classification setting (a false positive negates financial services to a legitimate applicant), we aim to balance false positive rates between the different groups in the dataset.

As a protected attribute for this dataset, we selected the applicant's age. Specifically, we divide applicants into two groups: under 50 years old, and at or above 50 years old. There is a surprising but clear relation between fraudulent applications and reported age, so we expect that fairness w.r.t. the age group will make for a challenging setting. Figure A3b shows a plot of the age distribution, as well as the variation of fraud rate over age values.

## C   DISCUSSION ON THE LIMITATIONS OF FAIRGBM

In this section, we discuss expected limitations of the proposed approach.

When compared with vanilla LightGBM, FairGBM requires approximately double the training time (see Table 2). This compute overhead of FairGBM can be attributed to (1) the computation of the proxy gradients relative to the constraint loss, and (2) the addition of an extra optimization step per iteration – the ascent step.

From an application standpoint, FairGBM is specifically tailored for gradient boosting methods. This contrasts with other bias mitigation methods in the literature such as EG (Agarwal et al., 2018), which, despite having considerably slower runtime, are applicable to a wider range of classifier types (all binary classifiers in the case of EG).

Despite discussing several choices for differentiable proxies for the Type 1 and Type 2 errors in Section 2.2, our experiments only concern one of these proxies. As future work, we would like to perform a more thorough study on the impact of different choices of differentiable proxies for the step-wise function (*e.g.*, sigmoid, hinge, or squared loss).

Moreover, FairGBM as it is devised in this work is limited to group-wise fairness metrics, and incompatible with metrics from the individual fairness literature. In fact, individual fairness algorithms tailored for GBM have been developed in related work Vargo et al. (2021). The BuDRO method proposed by Vargo *et al.* is based on optimal transport optimization with Wasserstein distances, while

our proposal fully relies on gradient-based optimization with the dual ascent method. A possible extension of our method to a wider range of constraints that enables individual fairness is a topic we aim to explore in future work.

## D OPERATING FAIRGBM AT A SPECIFIC ROC POINT

Without fairness constraints, meeting a specific ROC operating point can be achieved by appropriately thresholding the output of a classifier that learns to approximate the class probabilities: $p(y = 1|x)$ (Tong, 2013). That is the case of a classifier that is trained to minimize a proper scoring rule such as binary cross-entropy.

However, when optimizing for the Lagrangian (Equation 7), we are no longer optimizing for a classifier that approximates the true class probabilities. This is a key point that is often overlooked in the constrained optimization literature. Namely, both $\mathcal{L}$ and $\tilde{\mathcal{L}}$ have an implicit threshold when evaluating constraint fulfillment: the $0.5$ decision threshold, or $0.0$ when considering the log-odds (see Figure 1). In practice, FairGBM will be optimized to fulfill the constraints at this pre-defined threshold, but constraint fulfillment may not (and likely will not) generalize to all thresholds. Indeed, we could use any decision threshold during training, but it is impossible to know which threshold would meet our ROC requirements beforehand.

We propose to solve this by introducing our ROC requirement as another in-training constraint. In the AOF setting, this is achieved by introducing an additional constraint of global $FPR \leq 0.05$. In practice, instead of shifting the threshold to meet our target ROC operating point, we are shifting the score distribution such that the $0.5$ decision threshold corresponds to our target ROC point, enabling constraint fulfillment at any attainable ROC point.

## E RANDOMIZED CLASSIFIER *vs* LAST ITERATE

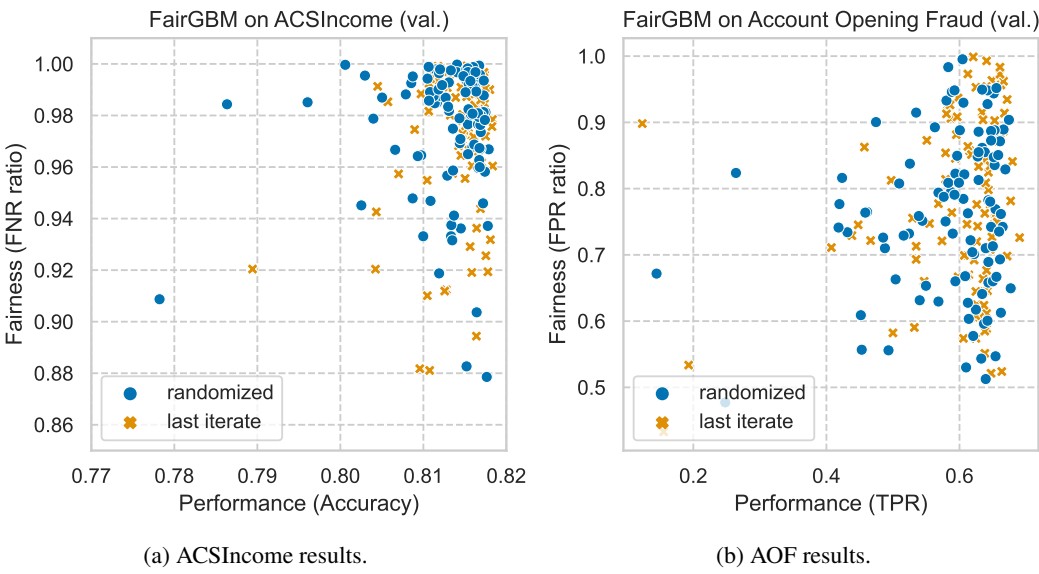

(a) ACSIncome results.          (b) AOF results.

Figure A4: Comparison between using the FairGBM randomized classifier (blue circles) or the predictions of the last FairGBM iterate (orange crosses).

Figure A4 shows a comparison between using the randomized classifier predictions or solely the last iterate predictions on the ACSIncome dataset. As mentioned in Section 2, the first approach benefits from theoretical convergence guarantees, while the latter benefits from being a deterministic classifier (which may be a requirement in some real-world settings). In practice, using the last iterate version of FairGBM (which always uses all trees of the GBM ensemble) achieves similar results to the randomized classifier version (which randomly picks the number of trees it will use for each

prediction). The same trend is clear on the AOF dataset, and concurs with related work on randomized classifiers by Cotter et al. (2019b).

# F BACKGROUND ON GRADIENT BOOSTED DECISION TREES

A gradient boosting algorithm estimates a mapping $f : \mathbf{X} \mapsto \mathbf{y}$ that minimizes a loss function,

$$L(f) = \frac{1}{N} \sum_{i=1}^{N} l(y_i, f(x_i)), \tag{10}$$

where f is constrained to be a sum of base (weak) learners $h_t \in \mathcal{H}$. In the case of GBDT, these can be shallow decision trees with fixed depth or fixed number of nodes:

$$f = \sum_{t=0}^{T} \eta_t h_t, \tag{11}$$

where $\eta_t$ is a step size parameter. Typically, $h_0$ would be a constant function that minimizes $L$ and $\eta_0 = 1$. Gradient boosting can then be understood as performing gradient descent on the space of functions $f$. Each subsequent step, $h_t$, being essentially a projection onto $\mathcal{H}$ of the negative gradient of the loss $L$ w.r.t. $f$. In other words, the base learner whose predictions are as close as possible, in the $l_2$ sense, to the negative gradient[2]:

$$h_t = \underset{h \in \mathcal{H}}{\arg\min} \sum_{i=1}^{N} \left( -g_{t,i} - h(x_i) \right)^2, \tag{12}$$

where $g_{t,i}$ are the gradients evaluated at the current iterate $f_{t-1} = \sum_{m=0}^{t-1} \eta_m h_m$:

$$g_{t,i} = \left[ \frac{\partial l(y_i, f(x_i))}{\partial f(x_i)} \right]_{f(x_i) = f_{t-1}(x_i)}. \tag{13}$$

Note that Equation 12 is equivalent to:

$$h_t = \underset{h_t \in \mathcal{H}}{\arg\min} \sum_{i=1}^{N} \left[ g_{t,i} h_t(x_i) + \frac{1}{2} h_t^2(x_i) \right]. \tag{14}$$

XGBoost (Chen & Guestrin, 2016) and LightGBM (Ke et al., 2017) replace the approximation above with a local quadratic one thus implementing the following second order step:

$$h_t = \underset{h_t \in \mathcal{H}}{\arg\min} \sum_{i=1}^{N} \left[ g_{t,i} h_t(x_i) + \frac{1}{2} H_{i,t} h_t^2(x_i) \right] + \Omega(h_t), \tag{15}$$

where $H_{i,t}$ is the hessian of $l$ w.r.t. $f$ computed at the current iterate and $\Omega$ is a regularization term penalizing complex base learners.

---

[2]In practice, a heuristic is used that builds the decision tree by greedily choosing a sequence of splitting variables and splitting values that most decrease the value of the function to minimize

## G  NOTATION

| | |
|---|---|
| $\mathcal{L}$ | the Lagrangian function, which uses the original constraints $c$; see Equation 2. |
| $\tilde{\mathcal{L}}$ | the proxy-Lagrangian function, which uses the proxy constraints $\tilde{c}$; see Equation 7. |
| $c$ | an inequality constraint function; it is deemed fulfilled if $c(\theta) \leq 0$; this function may be non-differentiable; examples include a constraint on TPR parity or parity of any other metric of the confusion matrix. |
| $\tilde{c}$ | a proxy inequality constraint that serves as sub-differentiable proxy for the corresponding constraint $c$; see Equation 5. |
| $l$ | an instance-wise loss function, i.e., $l : \mathcal{Y} \times \hat{\mathcal{Y}} \mapsto \mathbb{R}_+$, where $\mathcal{Y}$ is the set of possible labels and $\hat{\mathcal{Y}}$ is the set of possible predictions; see green line in Figure 1. |
| $\tilde{l}$ | a sub-differentiable proxy for an instance-wise loss function; see blue and purple lines in Figure 1. |
| $D$ | a dataset of samples $(x, y, s) \in D$, where $x \in \mathcal{X} \subseteq \mathbb{R}^n$ is the features, $y \in \mathcal{Y} \subseteq \mathbb{N}_0$ is the label, and $s \in \mathcal{S} \subseteq \mathbb{N}_0$ is the sensitive attribute. |
| $L_{(S=a)}$ | a predictive loss function measured over data samples with sensitive attribute value $S = a$, $\{(x, y, s) : s = a, (x, y, s) \in D\}$; the subscript is omitted when measuring loss over the whole dataset $D$; examples include the false-negative rate or the squared error loss. |
| $\tilde{L}_{(S=a)}$ | a sub-differentiable proxy for a predictive loss function measured over data samples with sensitive attribute value $S = a$, $\{(x, y, s) : s = a, (x, y, s) \in D\}$; the subscript is omitted when measuring loss over the whole dataset $D$. |
| $\lambda_i$ | a Lagrange multiplier associated with constraint $c_i$ and proxy constraint $\tilde{c}_i$. |
| $\mathcal{F}$ | the space of strong learners. |
| $\mathcal{H}$ | the space of weak learners. |
| $f$ | a strong learner. |
| $h$ | a weak learner. |
| $\mathcal{S}$ | the range of random variable $S$; the letter $S$ specifically is used for the sensitive attribute, and $\mathcal{S}$ for the different values the sensitive attribute can take. |

## H  REPRODUCIBILITY CHECKLIST

This section provides further details regarding implementation and hardware used for our experiments. We follow the reproducibility checklist put forth by Dodge et al. (Dodge et al., 2019).

Regarding reported experimental results:

☑ Description of computing infrastructure.
  - ACSIncome and AOF experiments: Intel i7-8650U CPU, 32GB RAM.
  - ACSEmployment, ACSMobility, ACSTravelTime, ACSPublicCoverage experiments: each model trained in parallel on a cluster. Resources per training job: 1 vCPU core (Intel Xeon E5-2695), 8GB RAM[3].

☑ Average run-time for each approach.
  - Folder `runtimes` of the supp. materials[4].

☑ Details of train/validation/test splits.
  - See Section 3.1, and data notebooks in folder `notebooks` of the supp. materials[4].

☑ Corresponding validation performance for each reported test result.
  - Folder `results` of the supp. materials[4].

☑ A link to implemented code[1].

Regarding hyperparameter search:

---

[3]These experiments were added as part of the paper rebuttal process, and thus required faster infrastructure to meet the conference deadlines.

[4]https://github.com/feedzai/fairgbm/tree/supp-materials

☑ Bounds for each hyperparameter.
- Folder `hyperparameters` of the supp. materials[4].

☑ Hyperparameter configurations for best-performing models.
- Folder `hyperparameters` of the supp. materials[4].

☑ Number of hyperparameter search trials.
- LightGBM, FairGBM, and RS: 100 trials.
- EG and GS: 10 trials — each trial trains $n = 10$ separate base models, for a total budget of 100 models trained (equal budget for all algorithms).

☑ The method of choosing hyperparameter values.
- Random uniform sampling.

☑ Expected validation performance.
- Folder `others` of the supp. materials[4].

