# OpenReview forum: "FairGBM: Gradient Boosting with Fairness Constraints"
_ICLR.cc/2023/Conference — ICLR 2023 poster_

### Official Review · Reviewer_MVBV · 2022-10-21

**Confidence:** 5
**Correctness:** 3
**Technical Novelty And Significance:** 2
**Empirical Novelty And Significance:** 2
**Recommendation:** 5

**Clarity, Quality, Novelty And Reproducibility:**

The paper is overall well-written. It's clear and easy to understand. Key resources (e.g., proofs, code, data) for reproducibility are available with sufficient details. But the novelty of the paper is limited.

**Strength And Weaknesses:**

Strength:
- The paper is well written and well motivated.
- There are not much existing work that focused on fairness for GBDT models.
- In depth formalization of the proposed framework

Weakness:
- It is not clear how the proxy-Lagrangian can be designed/derived for any fairness metrics. Is the cross-entropy-based proxy only applicable for demographic parity, equal opportunity and predictive equality?
- The proposed constrained optimization method works for GBDT-based classifier. Is it possible to address unfairness in regression task with the current approach? What would be the proxy function on that case? Note that Agarwal et al., 2019 extended their reduction-based approach for regression task as well. It would be great if the authors include discussion or additional experiments on this.
- Baseline models EG, GS and RS were trained with 10 iterations for an equal budget for all algorithms. But Agarwal et al., 2018 stated that 5 iterations were enough for their reduction approach in all cases. Additional experiments with varying number of iterations for all models would be helpful to understand if EG were not overfitted with additional iterations.
- Instead of completely discarding EG from the experiment of real-world data, why not changing the decision threshold and n in EG for feasibility, in case of this dataset.
- How trade-off \alpha = 0.5, 0.75, and 0.95 selected? Fairness-accuracy trade-off plots for all models would be helpful to understand the impact of different constraint methods.
- Fairlearn package is mainly implemented in Python, while FairGBM is implemented in C++. Does FairGBM get additional runtime benefit due to C++?


**Summary Of The Paper:**

This paper is focused on ensuring fairness on tabular data. Since Gradient boosted decision trees (GBDT) - based classifiers are found effective on tabular data, the authors propose FairGBM framework for training GBDT under fairness constraints. The experimental result shows improvement in the training time, predictive performance and fairness.

**Summary Of The Review:**

Although the authors claimed to develop a learning framework for fair GBDT, the current approach is only applicable for classification task. Furthermore, constrained optimization is not new in fair ML research. Their formulation of constrained optimization is very similar with  (Agarwal et al., 2018)'s reduction approach. The only difference is using the differentiable proxy functions which was adapted from (Cotter et al. 2019). So, the main ideas of the paper have limited novelty. In addition, the paper lacks adequate experiments to establish author's claim. See above section.

-------------------------------------------------
I thank the authors for explaining some of my concerns. Although I am not completely convinced regarding the novelty of the paper, after reading other reviewer's comments, I have decided to increase my rating.

---

> ### Author Response · Authors · 2022-11-19
> **Response to Reviewer MVBV**
>
> Thank you for the thorough review. We are going to address the raised concerns in separate comments.

---

> ### Author Response · Authors · 2022-11-19
> **Response on the generality of constraints compatible with FairGBM**
>
> > It is not clear how the proxy-Lagrangian can be designed/derived for any fairness metrics. Is the cross-entropy-based proxy only applicable for demographic parity, equal opportunity and predictive equality?
>
> We define the cross-entropy-based proxy loss for every possible model outcome, i.e., for a given real-value score prediction $f(x)$, we define the proxy loss for label negatives (Table 1, first row), or label positives (Table 1, second row); when considering the confusion matrix, these errors correspond to false positives or false negatives, respectively. Hence, any function of the confusion matrix (or of the confusion matrices of different population groups) can be modeled using cross-entropy-based proxies. In this way, we provide general building blocks for any fairness metric that fulfills the aforementioned description, which includes the most popular metrics in the Fair ML literature, such as, equal odds (equal TPR and TNR), equal opportunity (equal TPR), predictive equality (equal TNR), and demographic parity (equal positive prediction rate). Nonetheless, there are bound to exist more fairness metrics in the literature that fit this description.

---

> ### Author Response · Authors · 2022-11-19
> **Response on the use of FairGBM for regression tasks**
>
> > The proposed constrained optimization method works for GBDT-based classifier. Is it possible to address unfairness in regression task with the current approach? What would be the proxy function on that case? Note that Agarwal et al., 2019 extended their reduction-based approach for regression task as well. It would be great if the authors include discussion or additional experiments on this.
>
> Although the proposed constrained optimization method was formalized for classification, it could also be used for regression. In fact, the only significant difference would be the choice of fairness metric. Unlike classification, in a regression setting there would likely be no need for a proxy metric, as common regression fairness metrics are differentiable and can be directly optimized (for example, those used in [Agarwal et al. (2019)](http://proceedings.mlr.press/v97/agarwal19d/agarwal19d.pdf)).
>
> Fairness metrics generally target equality between group-wise performance metrics [(Barocas et al., 2019)](https://fairmlbook.org/pdf/fairmlbook.pdf). In classification, as explored in Section 2, this leads to the necessity of using differentiable proxies for these performance metrics; on the other hand, regression expectedly does not suffer from this same problem, as group-wise performance metrics would be differentiable. Nonetheless, it depends on the specific fairness metric chosen.
>
> To conclude, while the application of our method to regression settings is straight-forward, in this paper we focus on the classification setting. We have added a sentence to the paper’s body bringing attention to this distinction. Using and benchmarking FairGBM on datasets for regression tasks is definitely interesting for future work.

---

> ### Author Response · Authors · 2022-11-19
> **Response on the use and training of EG models**
>
> > Baseline models EG, GS and RS were trained with 10 iterations for an equal budget for all algorithms. But Agarwal et al., 2018 stated that 5 iterations were enough for their reduction approach in all cases.
>
> Thank you for raising this point. The default value for the number of iterations with EG in the fairlearn package is n=50; we reduced this to n=10 because (1) as pointed out, the authors claim n=5 should be enough in most cases, and (2) to be able to get more data points from the EG method, as using n=50 would lead to training only two EG models.
>
> Another important point is that the datasets used in Agarwal et al. (2018) were all considerably smaller than the datasets we use in our experiments. In fact, the largest dataset used by Agarwal et al. is approximately 10x smaller than the smallest dataset used in our experiments, with 49K samples and 500K samples respectively.
>
> Indeed, we could also consider that a fairer comparison would be to use CPU time as a  budget, instead of simply using the number of training trials. Using CPU time as a budget would lead to a much higher advantage to FairGBM, and worse results for EG and the other baselines, as FairGBM has a lower CPU consumption per training trial. Conversely, by using the number of training trials as the budget unit we are in fact benefiting EG and the remaining baselines over our proposed method.
>
> All in all, we believe we gave all baselines a worthy opportunity, as in fact FairGBM had the least CPU budget of all and still managed to outperform in most aspects.

---

> ### Author Response · Authors · 2022-11-19
> **Response on the incompatibility of EG with the AOF task**
>
> > Instead of completely discarding EG from the experiment of real-world data, why not changing the decision threshold and n in EG for feasibility, in case of this dataset.
>
> Thank you for raising this point. The incompatibility of EG with the AOF real-world task is precisely tied to the fact that it is limited to binary predictions, i.e., it only has a single possible decision threshold (Agarwal et al., 2018).
>
> This limitation of EG leads to incompatibility with requirements that are commonly present in real-world automated decision making systems, e.g., a limited number of accepted/denied applicants (limit on the number of positive/negative predictions), or a limited number of false positives (as these can be expensive and/or increase customer attrition). One way to fulfill these requirements would be to randomly untie samples with the same prediction (of which there will be many as there are only two possible predictions $\hat{y} \in \{0, 1\}$). We think this would not make a fair comparison, so we instead use EG on the benchmark dataset (ACSIncome) and highlight its incompatibility with the AOF real-world setting.

---

> ### Author Response · Authors · 2022-11-19
> **Response on results for other fairness-performance trade-offs**
>
> > How trade-off \alpha = 0.5, 0.75, and 0.95 selected? Fairness-accuracy trade-off plots for all models would be helpful to understand the impact of different constraint methods.
>
> The full 2-D fairness accuracy plot can be seen in Figure 2.
>
> As a result of your feedback we have decided to add new plots on the $\alpha$-weighted metric for a clearer comparison between different models under this metric (Figures 4, 5, 6). These plots provide a different continuous visualization of the attainable fairness-accuracy trade-offs for each model. The conclusions of the experiments section are largely maintained.
>
> Thank you for this suggestion.

---

> ### Author Response · Authors · 2022-11-19
> **Response on the use of C++ or Python code for fairlearn**
>
> > Fairlearn package is mainly implemented in Python, while FairGBM is implemented in C++. Does FairGBM get additional runtime benefit due to C++?
>
> While the _fairlearn_ package is implemented in Python, the high compute requirements of EG/GS/RS arguably stem from training the underlying model many times. As the underlying model is LightGBM (a highly efficient C++ GBM implementation by Microsoft), all baselines will perform the most computationally heavy tasks - training the ML model - in compiled C++ code as well as FairGBM.
>
> At the same time, we too provide a Python package for _fairgbm_, to foster wider adoption of the proposed algorithm, as Python libraries are arguably easier to use than C++ libraries. This _fairgbm_ package then calls the C++ implementation for the most computationally model training, as well as for other not so heavy tasks such as computing predictions, computing global and group-wise metrics. All in all, we use Python interfaces to train and evaluate all models in our experiments, but training for all models runs in binaries compiled from C++ code.

---

> ### Author Response · Authors · 2022-11-19
> **We thank the reviewer for the insightful recommendations**
>
> We thank the reviewer for the insightful recommendations. Based on our comments and modifications to the paper we sincerely encourage the reviewer to revise the awarded rating. Thank you once again for the thorough review.

---

### Official Review · Reviewer_Cc3g · 2022-10-24

**Confidence:** 4
**Correctness:** 3
**Technical Novelty And Significance:** 2
**Empirical Novelty And Significance:** 3
**Recommendation:** 6

**Clarity, Quality, Novelty And Reproducibility:**

AOF dataset needs improved documentation/discussion, if it is not open source. Otherwise, the paper provides an open-source implementation which appears to be an important contribution of the work, and should greatly aid in reproducibility.

**Strength And Weaknesses:**

## Major Comments

* The authors note that there are clear differences between the two datasets (e.g. differences in label imbalance, differences in spread over performance metrics). Given this, it is hard to make much sense of the results, since they seem to imply meaningfully different conclusions (i.e. Fairlearn EG achieves excellent tradeoffs on ACS Income and much lower variance than the proposed method; GS and RS achieve near-perfect fairness on AOF, but not on ACSIncome). Given that the authors are already (I assume) using the folktables package to generate ACS Income dataset, why not use additional tasks already available in that package (Public Coverage, etc.)? This would also provide more publicly verifiable results on open datasets.

* The AOF dataset is not clearly described. Is it an open dataset (I do not see it in supplement)? What is the source? What features exist in the dataset, what are their data types, etc.? Particularly if the dataset is *not* open, much more detail would be useful.

* Given the very strong performance of EG on ACSIncome, I wonder why more effort was not made to include it in AOF experiments. If EG gives a randomized binary classifier, why can't the average of many (randomized) predictions be used as a de facto continuous prediction?

* The authors perform 10 experimental iterates, but do not use these iterates to provide measures of statistical variability in comparing their point estimates -- a missed opportunity, and also a critical one given that several similar algorithms are being compared. Please provide, if possible, estimates of variation (i.e. SD over trials, Clopper-Pearson confidence intervals, etc.) of the various metrics being compared e.g. in Table 2, Figure 2. This will help the authors eliminate subjective language in their analysis such as "excellent trade-offs", "only slighting lagging behind", etc.

* The paper is missing a discussion of the limitations of the proposed approach. Please comment on this.

* It's not clear why equalizing the group FNR make sense for any real-world task on ACS Income. Please comment on the motivation for this, or choose a constraint that actually has some grounding in the income task.

* I think the paper should comment on, and ideally compare to, the approach described in "Individually Fair Gradient Boosting", Vargo et al., https://arxiv.org/pdf/2103.16785.pdf ).

## Minor Comments

* Please use some other form of marker to differentiate between models in Fig. 2; it is difficult to see the differenecs between colors.

* Hwo is the alpha=0.75 in Table 2 chosen? If it is arbitrary, how do the comparisons change with \alpha? Is it possible to instead plot curves over all values of alpha in [0,1]?

## Typos etc

P1 "no clear winner method" -> no clear winning method

Please spell out "w.r.t" in the paper

Equation (5): If I understand correctly, the max can be over a,b \in S (eliminating the need for \forall b \in S)

I don't see S_s or L_S_s defined in the paper.

P8: "significative"

**Summary Of The Paper:**

The paper presents a method for training gradient-boosted trees with fairness constraints via a (proxy) Lagrangian approach. They provide an algorithm, an open-source implementation, and empirical results on two datasets (only one appears to be open-source) showing that the proposed algorithm generally improves over existing methods.

**Summary Of The Review:**

Overall, this paper pursues an important direction for machine learning research -- tree-based models are, as the authors note, SOTA for tabular data yet are incompatible with many gradient-based regularization approaches, so new methods are needed. In general, I think the paper is well-written, but the discussion of the experimental results could be improved. Given the clear differences between the two datasets, I also think that adding additional datasets (e.g. additional tasks from folktables) would considerably improve the paper. See detailed comments above/below.

---

> ### Author Response · Authors · 2022-11-19
> **Response to Reviewer Cc3g**
>
> Thank you for your review and thoughtful comments and suggestions. We have updated our paper based on your feedback and we will also address each point raised in the following separate comments.

---

> ### Author Response · Authors · 2022-11-19
> **Response on the clarification of results and conclusions**
>
> > The authors note that there are clear differences between the two datasets (e.g. differences in label imbalance, differences in spread over performance metrics). Given this, it is hard to make much sense of the results, since they seem to imply meaningfully different conclusions (i.e. Fairlearn EG achieves excellent tradeoffs on ACS Income and much lower variance than the proposed method; GS and RS achieve near-perfect fairness on AOF, but not on ACSIncome).
>
> Thank you for raising this point. The results achieved by EG on the ACSIncome dataset do indeed lag behind FairGBM in constraint fulfillment. We recognize this was not fully apparent in the submitted version of the paper, so we **added statistical variance results to the latest paper revision** (Table 2, 3, and 4), as suggested by the reviewer. Separate hyperparameter optimization (HPO) runs were obtained via bootstrapping (with 1000 repeated HPO trials; each trial consists of $n=20$ random draws from the pool of $100$ trained models, or $n=2$ from the pool of $10$ models for EG and GS). We have also added plots for the attainable fairness-performance trade-off of each model, for all $\alpha \in [0, 1]$ (Figures 4 and 5, with 95% confidence intervals in shade).
>
> With this new data, we can confirm that FairGBM achieves a superior fairness-performance trade-off to all baselines for all $\alpha \in [0.05, 0.98]$ in the AOF dataset (see Figure 4b), and for all $\alpha \in [0.00, 0.99]$ in the ACSIncome dataset (see Figure 4a). Indeed, on the ACSIncome, we can only consider EG results better than FairGBM by almost disregarding fairness (giving it a weight of under 1% in the model selection metric).
>
> Moreover, we can also see that both EG and FairGBM have very low variance in their results (Tables 2-4). As the outcome of the HPO process is choosing the maximum among all trained models, the expected HPO outcome is a mean over maxima, and the variance is computed over the maximum of each HPO run. Even though FairGBM results clearly have a wider spread on the fairness-performance plot of Figure 2 (when compared with EG), this spread is not indicative of the variance of outcomes we can expect after the HPO process and respective model selection (Tables 2-4).
>
> We recognize the previous data presentation was not ideal, and hope the new results with statistical variance results contribute to improve clarity of the results section.

---

> ### Author Response · Authors · 2022-11-19
> **Response on running FairGBM on extra benchmark datasets**
>
> > Given that the authors are already (I assume) using the folktables package to generate ACS Income dataset, why not use additional tasks already available in that package (Public Coverage, etc.)? This would also provide more publicly verifiable results on open datasets.
>
> Thank you for the suggestion of using the wider selection of datasets available on the folktables package. We agree that this will make the experimental section more complete.
>
> As pointed out in Table 2, running EG on ACSIncome took over 99 hours, and the remaining methods took another 95 hours in aggregate. As the remaining folktables datasets have a similar order of magnitude as ACSIncome, we unfortunately won’t be able to complete extra experiments until the rebuttal deadline (18th November). Nonetheless, **we commit to adding these experiments in the following weeks, to be included in the final version of the paper**.

---

> ### Author Response · Authors · 2022-11-19
> **Response on details for the AOF dataset**
>
> > The AOF dataset is not clearly described. Is it an open dataset (I do not see it in supplement)? What is the source? What features exist in the dataset, what are their data types, etc.? Particularly if the dataset is not open, much more detail would be useful.
>
> Thank you for raising this concern. To clarify, the AOF dataset is not open to the public. We believe that testing the proposed method and implementation in a real-world scenario is a relevant contribution, as it carries significant challenges that are not often present in public benchmark datasets (e.g., significant class imbalance of the dataset, and specific requirements on the allowed FPR of the model). For example, while simply minimizing cross-entropy optimizes for a calibrated classifier, this is no longer the case when introducing in-training fairness constraints (see discussion in Appendix C - “Operating FairGBM at a specific ROC point”). Given that AOF is not public, we agree that more detail should be provided, and we added a new dedicated Appendix (F) for this.

---

> ### Author Response · Authors · 2022-11-19
> **Response on the incompatibility of EG with the AOF task**
>
> > Given the very strong performance of EG on ACSIncome, I wonder why more effort was not made to include it in AOF experiments. If EG gives a randomized binary classifier, why can't the average of many (randomized) predictions be used as a de facto continuous prediction?
>
> The idea of averaging iterate predictions to achieve some granularity in the score function of EG is an interesting idea, and we confess it did cross our minds. Nonetheless, we decided against it as this new method would not correspond to EG, and it would arguably be a highly unfair comparison (disadvantaging EG), for two main reasons:
> 1. As EG uses at most 10 model iterates in our experiments, the best case scenario would be having 11 thresholds, $t \in \{0.0, 0.1, ..., 0.9, 1.0\}$. As this deterministic EG would only output 1 of 11 possible prediction scores, there would inevitably be numerous ties between different samples (note that the AOF dataset has over 500K samples). These ties would have to be randomly resolved to achieve the 5% FPR requirement of the AOF task, inevitably lowering the method’s performance.
> 2. The convergence to a $v$-approximate saddle point of the Lagrangian only happens for the randomized binary classifier (Agarwal et al., 2018); this property is not trivially attributed to the mentioned deterministic and continuous version of the method.
> Moreover, from an implementation stand-point, the predictions of each separate EG iterate are not readily available in the `fairlearn` package.
>
> Alternatively, we will address this concern by evaluating EG (together with FairGBM and the remaining baselines) on other benchmark datasets in a way that is compatible with all methods. As such, we will add extra EG results to the final version of the paper. Nonetheless, we think it is important to point out that EG is not easily compatible with the presented real-world setting (whose requirements are common among real-world tabular ML tasks).

---

> ### Author Response · Authors · 2022-11-19
> **Response on the addition of measures of statistical variance**
>
> > The authors perform 10 experimental iterates, but do not use these iterates to provide measures of statistical variability in comparing their point estimates -- a missed opportunity, and also a critical one given that several similar algorithms are being compared. Please provide, if possible, estimates of variation (i.e. SD over trials, Clopper-Pearson confidence intervals, etc.) of the various metrics being compared e.g. in Table 2, Figure 2. This will help the authors eliminate subjective language in their analysis such as "excellent trade-offs", "only slighting lagging behind", etc.
>
> We agree that measures of statistical variance would be a significant addition and we have added them to the paper. We will also carefully address this concern in the following paragraphs.
>
> As the result of the hyperparameter optimization (HPO) process is selecting the configuration with maximal result of the $\alpha$-weighted metric (i.e., $\alpha * performance + (1-\alpha) * fairness$) among all sampled configurations, what we would like to estimate is the expected _maximum_ result among $k$ random trials instead of the expected (_mean_) result among $k$ random trials.
>
> A naive approach to obtaining metrics of statistical variance would require running the whole HPO procedure several times (and computing the average and variance among the obtained maxima). As the experiments on both datasets totaled 231.4 hours, repeating the experiment $n=5$ times would take over 1000 hours.
>
> There have been proposals in the HPO literature for less naive approaches to estimating the expected maximum, one of the most well-known being that of [Dodge et al. (2019)](https://aclanthology.org/D19-1224.pdf). However, while this approach works fine for traditional HPO based on 1-D model selection (e.g., selecting the model that maximizes accuracy among all trained hyperparameter configurations), it is _not compatible_ with 2-D model selection (which is our case, selecting based on both performance and fairness). We could apply this method for the expected maximum on the $\alpha$-weighted metric, but not for the expected performance or fairness of the model that maximizes the $\alpha$-weighted metric.
>
> As such, we will use the popular Bootstrap method [(Efron and Tibshirani, 1994)](https://cindy.informatik.uni-bremen.de/cosy/teaching/CM_2011/Eval3/pe_efron_93.pdf), also used by, e.g., [Lucic et al. (2018)](https://proceedings.neurips.cc/paper/7350-are-gans-created-equal-a-large-scale-study) and [Henderson et al. (2018)](https://ojs.aaai.org/index.php/AAAI/article/view/11694). For a given model type, this method consists of drawing $k$ trials (trained models) from a pool of $n$ total available trials (all trained models), with $k<n$. This process is repeated several times (let’s call each repetition a “run”), and the expected maximum and its variance is computed from the set of maximum values for each separate run.
>
> We have now computed the expected maximum using this method, together with statistical variance, and have **added these new results to Tables 2, 3, and 4** (as well as 95% confidence intervals to Figures 4, 5, and 6). Conclusions and discussion of the results are mostly unchanged, as new results confirm the previously obtained ones. We would like to thank the reviewer once again for raising this point and we believe this is a relevant addition to the paper.

---

> ### Author Response · Authors · 2022-11-19
> **Response on the addition of a Limitations section**
>
> > The paper is missing a discussion of the limitations of the proposed approach. Please comment on this.
>
> Thank you for pointing this out. We have added a limitations section as an Appendix (B) where we discuss both the limitations of the proposed approach and some future work. This section includes comments on the run-time efficiency of FairGBM, the applicability to different classes of ML models, as well as the selection of the different proxy functions.

---

> ### Author Response · Authors · 2022-11-19
> **Response on the choice of fairness metric for the ACSIncome dataset**
>
> > It's not clear why equalizing the group FNR make sense for any real-world task on ACS Income. Please comment on the motivation for this, or choose a constraint that actually has some grounding in the income task.
>
> Thank you for raising this point. Indeed, the choice of fairness metric for a specific ML task is highly dependent on the real-world context in which it operates, namely the different consequences for different types of errors (false positives or false negatives). This decision is often subjective, and requires careful consideration regarding 1st and 2nd order outcomes of the model’s decisions.
>
> The ACSIncome dataset is not directly related to any real-world decision-making process. From a real-world application standpoint, we can use information about people’s income to assist low-income individuals. Examples of assistance could be providing additional social security benefits, healthcare, or fellowships for their children. Under an assistive setting, failing to predict assistance needs for people in specific groups (i.e., predicting negative when the true label is “needs assistance” - a false negative) will disproportionately deny access to the assistive program, potentially amplifying disparities between people in the different groups. Under such real-world circumstances, we believe that equalizing FNR is a suitable fairness constraint.

---

> ### Author Response · Authors · 2022-11-19
> **Response on the comparison with individual fairness methods for GBM**
>
> > I think the paper should comment on, and ideally compare to, the approach described in "Individually Fair Gradient Boosting", Vargo et al., https://arxiv.org/pdf/2103.16785.pdf ).
>
> Thank you for bringing this reference to the discussion. While at a high-level, both FairGBM and the work of Vargo et al. (or BuDRO) are bias mitigation strategies tailored for Gradient Boosting, they focus on different types of fairness.
>
> In particular, our work focuses on group fairness (achieving parity in expectation across groups), whereas the latter focuses on individual fairness (making the same predictions for similar instances). FairGBM is not compatible with individual fairness and the approach taken by Vargo et al. is not compatible with group fairness goals. Moreover, due to the natural differences between individual fairness goals and group fairness goals, the approach taken by Vargo et al. is distinct from our approach. Namely, our approach is based on differentiable proxies for group-fairness metrics, and training the GBM ensemble with a dual ascent method; while BuDRO is based on optimal transport optimization with Wasserstein distances.
>
> As such, we initially thought that comparing our approach to an individual fairness method could lead to some confusion by the reader. We have now added a reference to this method in the section on limitations of FairGBM (Appendix B), to bring attention to the fact that other GBM-tailored methods should be used when targeting individual fairness.

---

> ### Author Response · Authors · 2022-11-19
> **Response on the use of different markers in Figure 2**
>
> > Please use some other form of marker to differentiate between models in Fig. 2; it is difficult to see the differences between colors.
>
> Thank you for bringing this to our attention, we’ve applied this change in the latest paper revision.

---

> ### Author Response · Authors · 2022-11-19
> **Response on the evaluation of other $\alpha$ trade-off values**
>
> > Hwo is the alpha=0.75 in Table 2 chosen? If it is arbitrary, how do the comparisons change with \alpha? Is it possible to instead plot curves over all values of alpha in [0,1]?
>
> Tables 3 and 4 (all information was previously gathered solely in Table 3 of the Appendix in the pre-revision version of the paper) show results for two other alpha values: $\alpha=0.50, \alpha=0.95$. The purpose of choosing a specific value of $\alpha$ is just to simplify comparison between models to a 1-D model selection process. In contrast, Figure 2 shows a comprehensive overview of the whole fairness-accuracy trade-off. To clarify, Figures 2, 4, 5, 6 and Tables 2, 3, 4 show different views of the same underlying results.
>
> Based on your comments, together with those of other reviewers, we’ve **added new plots for the fairness-accuracy trade-off as a continuous function of $\alpha \in [0, 1]$** (Appendix E, Figures 4, 5, and 6).
>
> Thank you for this suggestion.

---

> ### Author Response · Authors · 2022-11-19
> **We thank the reviewer for their insightful recommendations**
>
> We would like to thank the reviewer again for their thorough review and insightful recommendations. We believe this input improved the overall quality of the paper and sincerely encourage the reviewer to revise the awarded rating based on our updated version and clarifications.

---

### Official Review · Reviewer_WMdi · 2022-10-25

**Confidence:** 4
**Correctness:** 4
**Technical Novelty And Significance:** 3
**Empirical Novelty And Significance:** 3
**Recommendation:** 8

**Clarity, Quality, Novelty And Reproducibility:**

Clarity& Quality: The paper is of high quality in terms of arguments made towards choosing the opportunity, discussion of relevant background work, mathematical choices around constrained optimization and choice of L, evaluation setup of experiments and description of results.

Novelty: The paper is novel in proposing differentiable proxy functions for regular fairness metrics on the basis of cross entropy loss.


Reproducibility: All the materials including the algorithm, datasets, implementation code, experimental setup are clearly provided to ensure high reproducibility. They have followed the reproducibility checklist thoroughly and closely.

**Strength And Weaknesses:**

Strengths:
1. Strong summarization of relevant work, including pre processing, in processing and post processing and gap analysis to find area of opportunity for method development that can lead to highest impact i.e. there is a strong lack of a gold standard method that just works across the board and key identification of in-processing step as area of development.
2. The method proposed is efficient and does not require additional training or storage for keeping intermediate training states.
3. Method is applied to two diverse datasets to cover a variety of issues, bias, class imbalance, budget constraints on positive predictions etc., one of these datasets is benchmarked
4. The paper proposed mentions that the work is generalizable to any differentiable constraints, not just fairness constraints.
5. Comparison to other methods is fairly strong towards assesment of performance and fairness.


Weakness:
1. While the authors explicitly focus their attention to tabular data for evaluation of FairGBM, it would have been thoughtful to discuss potential impact of these methods on other structured/unstructured datasets as well (such as images, natural language etc). Even though the authors discuss there is no gold standard method that works regardless of data format or bias, it is then counterintuitive to focus the attention to just tabular data, given there are several other formats of data available.
2. Could be worth testing on more benchmark datasets to increase confidence coverage.
3. While the work is supposedly generalizable to additional constraints, apart from fairness constraints, it is not tested or validated for any other type of constraint.
4. While authors claim FairGMB can be applied to any GB algorithm, most comparisons to other constrained optimized methods are done by implementing FairGBM on LightGBM, while other methods also use LightGBM as base learners.
5. The authors can perhaps provide some commentary of how this work can be extended to other ML models, including deep learning, beyond just gradient boosting methods.

**Summary Of The Paper:**

The paper proposes a first of its kind in-processing learning framework FairGBM for training GBDT, without affecting its performance, constrained by fairness. The paper tries to advanced the area of FairML, limiting risks of unfair or biased ML systems and aims to establish a gold standard method. The authors provide evidence that FairGBM is an order of magnitude faster compared to existing related work such as LightGBM, RS Reweighing, Fairlearn GS and Fairlearn EG when tested against benchmark datasets and is superior to them in both fairness, performance and runtime. The authors discuss background related work in detail by providing an in-depth analysis of pre-processing, in processing and post processing methods and argue why introducing fairness during the in processing is more beneficial vs the other two.

**Summary Of The Review:**

The authors provide sufficient ground of reasoning why FairML is necessary in terms on risks of biases and discrimination affecting various ML work and how tabular data, an important format of information, important across various applications could be a strong opportunity for evaluation of the method proposed. The authors provide sufficient background around relevant work and argue correctly why the choice of opportunity is in processing and why constrained optimization options with fairness metrics and their non-convex properties lead to the development of the mentioned proxy functions. Further the work put forward in terms of experimental setup and comparison to other methods is fairly strong to justify why this method could be considered as a gold standard method for tabular data and for gradient boosting models. Although the scope is limited, it is still strong and hence I propose that this work should be accepted.

---

> ### Author Response · Authors · 2022-11-19
> **Response to Reviewer WMdi**
>
> Thank you for your thorough feedback. We have taken the comments on board to improve and clarify the paper. We will address raised concerns in the following separate comments.

---

> ### Author Response · Authors · 2022-11-19
> **Response on the applicability of FairGBM to other data formats**
>
> > While the authors explicitly focus their attention to tabular data for evaluation of FairGBM, it would have been thoughtful to discuss potential impact of these methods on other structured/unstructured datasets as well (such as images, natural language etc). Even though the authors discuss there is no gold standard method that works regardless of data format or bias, it is then counterintuitive to focus the attention to just tabular data, given there are several other formats of data available.
>
> Thank you for raising this point. We find that, in general, Deep Learning based models are state of the art for computer vision and natural language processing tasks, but for tabular data it has been shown that tree-based models, namely Gradient Boosted Decision Trees (GBDT) are very hard to beat ([Shwartz-Ziv et al., 2021](https://doi.org/10.1016/j.inffus.2021.11.011); [Grinsztajn et al., 2022](https://arxiv.org/pdf/2207.08815.pdf)).
>
> At the same time, there was a constrained optimization framework specifically for Deep Learning (TensorFlow Constrained Optimization - TFCO)  but it was lacking one constrained optimization framework tailored for GBDT (until FairGBM).  We bring up the fact that there is no clear “gold standard” method to motivate the fact that while TFCO may be suitable to be used in practice to promote fairness in computer vision and NLP tasks, it does not fulfill the need for constrained optimization for every data type, and therefore, a constrained optimization framework tailored to GBDT - the mainstream family of models used in tabular data – was glaringly missing.
>
> We assume the upper-bound performance of FairGBM to be similar to unconstrained GBDT, which is not as competitive as Deep Learning in computer vision and natural language. As such, we had not considered including other data formats in the evaluation of FairGBM as we expect it will not be competitive in these data format. Nonetheless, we will run an evaluation of FairGBM using other data formats and will add it to the final version of paper in case it achieves competitive results.

---

> ### Author Response · Authors · 2022-11-19
> **Response on the use of more benchmark datasets**
>
> > Could be worth testing on more benchmark datasets to increase confidence coverage.
>
> Thank you for this comment and we agree that results on more benchmark datasets would be a valuable addition to the paper. We are running extra experiments on the several benchmark datasets available through the folktables package (from which we extracted the ACSIncome dataset). As the ACSIncome experiment took over 194 hours to run (see last column of Table 2), and the remaining datasets are similarly sized, new results will expectedly take too long to be included before the November 18th rebuttal deadline. Nonetheless, we commit to complete these experiments in the following weeks, and include these results in the final version of the paper.

---

> ### Author Response · Authors · 2022-11-19
> **Response on testing other non-fairness constraints**
>
> > While the work is supposedly generalizable to additional constraints, apart from fairness constraints, it is not tested or validated for any other type of constraint.
>
> The focus of the present work is on fairness due to, among other aspects, the particular difficulties it presents (common fairness notions are non-convex and non-differentiable). We believe that testing our method on the fairness setting poses a wide set of challenges in common with other non-fairness constraints. The code for our implementation is (will be) open-source, and extra constraints can be easily introduced. We welcome the implementation of other types of constraints by the research community, and hope to ourselves contribute to this in future work.

---

> ### Author Response · Authors · 2022-11-19
> **Response on the use of FairGBM with other GBM implementations**
>
> > While authors claim FairGBM can be applied to any GB algorithm, most comparisons to other constrained optimized methods are done by implementing FairGBM on LightGBM, while other methods also use LightGBM as base learners.
>
> Indeed, the proposed method (described in Section 2) is compatible with any gradient boosting algorithm, and there are even no assumptions on the function space $\mathcal{F}$ used in Algorithm 1. In practice, we always need to instantiate this choice of algorithm in order to run FairGBM. With the goal of wide adoption of our package and implementation, we decided to develop it on top of the Microsoft LightGBM code base. LightGBM is a popular (199M pypi downloads as of today) and highly efficient GBM implementation. Even though it does provide slightly non-standard GBM features (such as exclusive feature bundling ([Ke et al., 2017](https://proceedings.neurips.cc/paper/2017/hash/6449f44a102fde848669bdd9eb6b76fa-Abstract.html))), you can also train a standard GBM by setting `boosting_type=gbdt` and `enable_bundle=False` and we expose this option in our implementation. We argue that the use of LightGBM is just an implementation detail, and is not expected to alter the results in any significant manner. The one aspect that is advantaged by using LightGBM is its fast runtime, but this is not a comparative advantage for FairGBM as all baselines also use the same highly-efficient LightGBM implementation. At the same time, as the FairGBM development effort consisted of approximately 7k C++ line additions over the standard LightGBM code base, replicating this effort for different GBM implementations would be a strenuous effort for this paper submission. Nonetheless, we will welcome contributions from the research community to our package, including implementation of FairGBM using other GBDT implementations (e.g., XGBoost).

---

> ### Author Response · Authors · 2022-11-19
> **Response on extension of the FairGBM algorithm to other underlying model types**
>
> > The authors can perhaps provide some commentary of how this work can be extended to other ML models, including deep learning, beyond just gradient boosting methods.
>
> Some of the assumptions that allow FairGBM to be more efficient than other constrained optimization methods require its training procedure to be based on gradient boosting; for example, with a boosting training procedure, each model iteration contains all of the previous iteration’s weak learners. In order to converge to approximate constraint fulfillment, we define FairGBM as a randomized classifier over all of the iterates of the training procedure, which (due to boosting) are easily accessible.
>
> A similar procedure can be applied to Deep Learning, but this requires keeping several model iterates in memory. A separate point in favor of gradient boosting methods is that the gradient descent step is performed with relation to a convex function, while in Deep Learning the loss is not a convex function of the model’s parameters; i.e., cross-entropy loss is a convex function of the model’s predictions, but it’s not a convex function of a neural network’s parameters. This Deep Learning dual ascent procedure has been explored by [Cotter et al. (2018)](https://www.jmlr.org/papers/volume20/18-616/18-616.pdf), and implemented in the [TFCO](https://github.com/google-research/tensorflow_constrained_optimization) framework.
>
> All in all, a general version of the proxy-Lagrangian procedure and the proposed differentiable proxy functions for popular fairness metrics based on the cross-entropy loss can be applied to any gradient-based optimization method.
>
> With this in mind, we would like to thank the reviewer again for their suggestions, and hope our responses cleared up any concern regarding our work.

---

### Official Review · Reviewer_f61N · 2022-10-26

**Confidence:** 3
**Clarity, Quality, Novelty And Reproducibility:** Overall, paper is good written and we…
**Correctness:** 3
**Technical Novelty And Significance:** 2
**Empirical Novelty And Significance:** 2
**Recommendation:** 6

**Strength And Weaknesses:**

Strength
[+] The problem is well motivated. Fairness is an important topics and designing a optimization framework in GBDT to ensure fairness is an interesting and important area.


Weakness
[-] FairGBM is a modified version of LightGBM with fairness constraint. From the experimental results in Table 2, it shows FairGBM is better than lightGBM in both performance and fairness. Could the author helps to clarify the source of improvement of performance? It seems contra intuitive that a fairness constraint algorithm outperforms a unconstraint algorithm that purely optimize for performance.

[-] Could the author helps to confirm fairness metrics used in evaluation and model training (\tiltle(c)_i in equation 7)? In figure 2, the fairness metrics is different across dataset, FNR for ACSIncome-Adult and FRR for Account Opening Fraud. Is different fairness constraint selected when algorithm is applied on different datasets?

**Summary Of The Paper:**

The paper proposed a new FairGBM method to train GBDT under fairness constraints that shows little impact to predictive performance but improved fairness metrics as compared to unconstrained GBDT. The major challenge in the Fairness constraint is from the non-differentiable property of fairness notion, which is resolved by the proxy Lagrangian in this paper. Afterwards, the problem is resolve under the two-player game formulation where a descent step optimizes the loss and a ascent step ensures the fairness. Numerical results on ACSIncome-Adult and AOF dataset show that FairGBM proposed from the paper have better trade-off among Fairness, Performance, and Efficiency.

**Summary Of The Review:**

Overall the paper proposed a fairGBM with a proxy Lagrangian formulation. The problem is well motivated and solution is reasonable sound. There are some doubts on the experimental results, while overall results demonstrate the efficiency of the new proposed algorithm.

---

> ### Author Response · Authors · 2022-11-19
> **Response to Reviewer f61N**
>
> Thank you for your careful analysis of our work. We are very glad to know that you appreciate the significance of our work. We will address specific questions in the following paragraphs.
>
> > From the experimental results in Table 2, it shows FairGBM is better than lightGBM in both performance and fairness. Could the author helps to clarify the source of improvement of performance? It seems contra intuitive that a fairness constraint algorithm outperforms a unconstraint algorithm that purely optimize for performance.
>
> It is, as you point out, counter-intuitive to expect a constrained method to perform better than its unconstrained counterpart. This is visible in Figure 2 for both datasets: LightGBM, in orange, achieves better results on the x-axis (performance) than FairGBM, in blue. This 2-dimensional plot represents all attainable fairness-accuracy trade-offs. On the other hand, Tables 2 and 3 show a common but simplified view over the same set of trained models: choosing a target trade-off (alpha) between the two metrics and reducing the problem to a single dimension, i.e., selecting the models that maximize $\alpha * performance + (1 - \alpha) * fairness$. If we were to select $\alpha=1.00$, i.e., only consider the performance metric, then LightGBM would show better performance results; but when striving for some fairness ($\alpha=0.75$ means 75% importance to performance and 25% importance to fairness) FairGBM shows better fairness *and* performance. For an intuitive explanation, this happens because LightGBM has to sacrifice more performance points for each fairness point gained, which is one of the major advantages of FairGBM.
>
> All in all, although the best-performing (ignoring fairness) model is a LightGBM model, there is a significant region of fairness-performance space in which FairGBM models are Pareto dominant over LightGBM models (you can obtain both fairness and performance improvements at the same time).
>
> We have added plots for the continuous fairness-performance trade-off in Appendix E. We hope this response addresses your concerns.
>
> ___
> > Could the author helps to confirm fairness metrics used in evaluation and model training (\tiltle(c)_i in equation 7)? In figure 2, the fairness metrics is different across dataset, FNR for ACSIncome-Adult and FRR for Account Opening Fraud. Is different fairness constraint selected when algorithm is applied on different datasets?
>
> The choice of fairness metric to use as constraint is usually dependent on the dataset and task at hand. This is common practice in Fair ML evaluation as depending on the application the cost of a false positive may be higher, lower or the same as a false negative.  A constraint on equality of FNR is used when training fairness-aware methods on the ACSIncome dataset, and a constraint on equality of FPR is used for the Account Opening Fraud dataset. We have now made this more explicit in the main paper. Thank you for pointing out the need for this clarification.
>
> ___
>
> We would like to thank the reviewer again for their thorough review and insightful recommendations. We believe the latest paper changes have considerably improved the presented work, and have hopefully cleared up any concerns. We sincerely encourage the reviewer to revise the awarded rating based on our updated paper revision and clarifications.

---

### Author Response · Authors · 2022-11-19
**Global summary of reviewer feedback and paper changes**

We thank the reviewers for taking the time to review our work and for the thoughtful constructive feedback that helped us to revise our paper according to recommendations and criticisms.

We are very glad that all reviewers found our paper well written and well motivated. Additionally, reviewers pointed out the strong formalization of our proposed framework (Reviewer MVBV) or the soundness of our solution (Reviewer f61N); the efficiency of our open-source implementation (Reviewers f61N, WMdi); and the scarcity of existing work on fairness for GBDT models (Reviewers Cc3g, MVBV), which are mainstream and generally the state-of-the-art for tabular data tasks.

The reviewers provided valuable feedback on possible improvements, with a general focus on the paper’s experiments section. Reviewer Cc3g points out that more details should be provided on the in-house AOF dataset, which we have addressed by providing the dataset’s details in Appendix F. Reviewer Cc3g also suggests adding plots for the attainable fairness-performance values as a continuous function of $\alpha$, which we have added in Appendix E; as well as adding statistical variance measures to the results in Tables 2 and 3 (also addressed and included in the latest paper revision). As per Reviewer f61N’s feedback, we have also updated the paper notation to clear up possible notation overload; including the distinction between the random variable $S$, its range, now represented as calligraphic $S$, $\mathcal{S}$, and the subset of samples with a specific sensitive attribute value, $S=s$, now represented as $D_{(S=s)}$. We believe this new notation is now consistent throughout the paper, but any further feedback is of course appreciated.

Moreover, Reviewers WMdi and Cc3g encourage us to provide experimental results on more datasets (besides the 2 datasets used for experiments). We agree that results on more benchmark datasets would be a valuable addition. We will be adding results on the remaining benchmark datasets from the folktables package [(Ding et al., 2021)](https://proceedings.neurips.cc/paper/2021/file/32e54441e6382a7fbacbbbaf3c450059-Paper.pdf), from which we already present results on the ACSIncome dataset. As results on the ACSIncome task took a total of 194.8 hours to run (see 2nd-to-last column of Table 2), results on similarly sized folktables datasets will expectedly take too long to be included before the November 18th rebuttal deadline. Nonetheless, we commit to add these additional results to the final version of the paper.
We addressed reviewers feedback by posting detailed individual responses below, but also making extensive edits both to the main paper and Appendices, namely:
- Addition of statistical significance of results in Tables 2, 3, and 4;
- Addition of new plots on the continuous fairness-accuracy trade-off as a function of $\alpha \in [0.0, 1.0]$ (Appendix E, Figures 4, 5, and 6);
- Description of AOF dataset is further detailed (Appendix F);
- Using a colorblind palette for all plots, and differentiating marker style per algorithm;
- Small notation change to avoid notation overload surrounding $S$;
    - $L_{(S=a)}$ now represents the performance metric $L$ measured over samples with a given sensitive attribute $S=a$;
    - $D_{(S=a)}$ now represents the set of training samples with a specific sensitive attribute $S=a$;
- Correction of minor typos.

We hope to have adequately addressed all the questions and comments raised by the reviewers; and we want to thank them once again for their time and effort in reviewing this paper. We will include all clarifications provided and additional results suggested in the final version of the paper.

---

### Decision · Program_Chairs · 2023-01-20

**Decision:**

Accept: poster

**Justification For Why Not Higher Score:**

N/A

**Justification For Why Not Lower Score:**

N/A

**Metareview: Summary, Strengths And Weaknesses:**

This paper presents FairGBM, a dual ascent learning framework for training GBDT under fairness constraints, with little to no impact on predictive performance compared to unconstrained GBDT. They employ a ``proxy-Lagrangian'' formulation using smooth convex error rate proxies to enable gradient-based optimization. They show an order of magnitude speedup in training time compared with related work.

++ The motivation is clear. The paper is clear. Comparison to other methods is fairly strong towards assessment of performance and fairness.

-- It seems incremental because the FairGBM is a modified version of LightGBM with a fairness constraint.

--  The discussion of the experimental results could be improved such as adding additional datasets (e.g. additional tasks from folktables) would improve the paper.

After carefully reading the paper, the reviews, and the author responses, the meta-reviewer believes this paper is sufficient to cross the bar of ICLR, and the authors did a good job in responding to reviewers' comments. The meta-reviewer recommends an acceptance.


**Note From Pc:**

if the above contains the word "oral" or "spotlight" please see: "oral" presentation means -> notable-top-5% and "spotlight" means -> notable-top-25%. As stated in our emails, we are disassociating presentation type from AC recommendations